



# Seasonality and response of ocean acidification and hypoxia to major environmental anomalies in the southern Salish Sea, North America (2014–2018)

Simone R. Alin[1], Jan A. Newton[2,3], Richard A. Feely[1], Samantha Siedlecki[4], and Dana Greeley[1]

[1] Pacific Marine Environmental Laboratory, National Oceanic and Atmospheric Administration, 7600 Sand Point Way NE, Seattle, Washington 98115, USA
[2] Applied Physics Laboratory, University of Washington, Box 355640, Seattle, Washington 98105, USA
[3] School of Oceanography, University of Washington, 1492 NE Boat St., Seattle, Washington 98105, USA
[4] Department of Marine Sciences, University of Connecticut, 1080 Shennecossett Road, Groton, Connecticut 06340, USA

**ORCID IDs**
Alin: 0000-0002-8283-1910
Newton: 0000-0002-2551-1830
Feely: 0000-0003-3245-3568
Siedlecki: 0000-0002-5662-7326
Greeley: 0000-0003-4356-5899

*Correspondence to*: Simone R. Alin (simone.r.alin@noaa.gov)

**Abstract.** Coastal and estuarine ecosystems fringing the North Pacific Ocean are particularly vulnerable to ocean acidification, hypoxia, and intense marine heatwaves as a result of interactions among natural and anthropogenic processes. Here we characterize variability during a seasonally resolved cruise time series in the southern Salish Sea (Puget Sound, Strait of Juan de Fuca) and nearby coastal waters for select physical (temperature, T; salinity, S) and biogeochemical (oxygen, $O_2$; carbon dioxide fugacity, $fCO_2$; aragonite saturation state, $\Omega_{arag}$) parameters. Medians for some parameters peaked (T, $\Omega_{arag}$) in surface waters in summer, while others (S, $O_2$, $fCO_2$) changed progressively across spring–fall, and all parameters changed monotonically or were relatively stable at depth. Ranges varied considerably for all parameters across basins within the study region, with stratified basins consistently the most variable. Strong environmental anomalies occurred during the time series, allowing us to also qualitatively assess how these anomalies affected seasonal patterns and interannual variability. The peak temperature anomaly associated with the 2013–2016 northeast Pacific marine heatwave–El Niño event was observed in boundary waters during the October 2014 cruise, but Puget Sound cruises revealed the largest temperature increases during 2015–2016 timeframe. The most extreme hypoxia and acidification measurements to date were recorded in Hood Canal (which consistently has the most extreme conditions) during the same period; however, they were shifted earlier in the year relative to previous events. During autumn 2017, after the heat anomaly, a distinct carbonate system anomaly with unprecedentedly



low $\Omega_{arag}$ and high $f\mathrm{CO_2}$ occurred in parts of the southern Salish Sea that are not normally so acidified. This novel "CO₂ storm" appears to have been driven by anomalous river discharge earlier in 2017, which resulted in enhanced stratification and inferred primary productivity anomalies, indicated by persistently and anomalously high $O_2$, low $f\mathrm{CO_2}$, and high chlorophyll.

Unusually, this $CO_2$ anomaly was decoupled from $O_2$ dynamics compared to past Salish Sea hypoxia and acidification events. The complex interplay of weather, hydrological, and circulation anomalies revealed distinct multiple stressor scenarios that will potentially affect regional ecosystems under a changing climate. Further, the frequencies at which Salish cruise observations crossed known or preliminary species sensitivity thresholds illustrates the relative risk landscape of temperature, hypoxia, and acidification anomalies in the southern Salish Sea in the present-day, with implications for how multiple stressors

may combine to present potential migration, survival, or physiological challenges to key regional species in the future. The Salish cruise data product used in this publication is available at https://doi.org/10.25921/zgk5-ep63 (Alin et al., 2022), with an additional data product including all calculated $CO_2$ system parameters available at https://www.ncei.noaa.gov/data/oceans/ncei/ocads/metadata/0283266.html (Alin et al., 2023b).

## 1 Introduction

Northeast Pacific Ocean ecosystems are particularly vulnerable to marine heatwaves, hypoxia, and ocean acidification—the increase in seawater carbon dioxide ($CO_2$) due to ocean uptake of anthropogenic $CO_2$ emissions, which drives declining pH and calcium carbonate saturation states ($\Omega$)—as a result of interactions among natural and anthropogenic processes. Located at the terminus of global oceanic thermohaline circulation, subsurface NE Pacific water masses have low oxygen ($O_2$) and high dissolved inorganic carbon (DIC) content resulting from respiratory processes during isolation from the atmosphere (e.g.,

Franco et al., 2021 and references therein). Naturally high NE Pacific $CO_2$ levels are enhanced further through the addition of anthropogenic $CO_2$ (Feely et al., 2004, 2016; Sabine et al., 2004). Eastern boundary current systems accentuate this vulnerability by bringing subsurface, naturally $O_2$-poor, $CO_2$-rich waters toward the surface through upwelling (Feely et al., 2008; Chavez and Messié, 2009; Chavez et al., 2017). Estuarine systems such as the Salish Sea are typically lower in buffering capacity and are already rich in $CO_2$ due to dynamic local biological, hydrological, and geochemical processes; this natural

estuarine acidification is amplified when oceanic waters acidified by the uptake of anthropogenic $CO_2$ are transported into the estuary via estuarine circulation (Feely et al., 2010; Wallace et al., 2014; Pacella et al., 2018; Cai et al., 2021; Hunt et al., 2022). Thus, estuaries connected to upwelling coastal systems, particularly in the NE Pacific, receive naturally acidified, low-oxygen marine waters relative to those in other coastal regions (e.g., Windham-Myers et al., 2018). Continually rising $CO_2$ emissions and other climate change effects on coastal and estuarine processes are expected to increase the spatial and temporal

prevalence of acidified estuarine conditions (Pacella et al., 2018; Evans et al., 2019; Jarníková et al., 2022). Further, fjord-like estuaries with entrance sills, like Puget Sound and Hood Canal, retain some of the outgoing waters via mixing over the sills (known as reflux, e.g., MacCready et al., 2021), so anomalies tend to persist longer in these basins (Jackson et al., 2018).



Since 2007, carbonate system observations throughout the water column in coastal and estuarine NE Pacific ecosystems have proliferated, providing insight into the dynamics of ocean acidification parameters, including both measured (dissolved inorganic carbon, DIC; total alkalinity, TA; and sometimes pH on the total scale, $pH_T$) and calculated ($pH_T$; $CO_2$ partial pressure or fugacity, $pCO_2$ or $fCO_2$; and calcium carbonate saturation states, aragonite: $\Omega_{arag}$, calcite: $\Omega_{calc}$) variables (e.g., Feely et al., 2008, 2010; Alin et al., 2022, 2023a). Hood Canal, having long been known as a hotspot for hypoxia (defined here as oxygen levels below 62 µmol kg$^{-1}$ = 2.0 mg L$^{-1}$ = 1.5 mL L$^{-1}$) (Newton et al., 2007), was shown to have the most severe aragonite undersaturation ($\Omega_{arag}<1$) in the southern Salish Sea during the first direct carbonate system measurements (Feely et al., 2010). Subsequent observations showed aragonite undersaturation to be prevalent throughout most of the water column, most of the time in the northern Salish Sea as well (Ianson et al., 2016; Evans et al., 2019), with numerical models showing that pre-industrial Salish Sea chemistry predisposed it to rapid expansion of undersaturated conditions (Bednaršek et al., 2020a; Jarníková et al., 2022). Surface climatologies of carbonate chemistry in marine surface waters throughout Washington state revealed strong seasonal variability, with particularly high $fCO_2$ and low pH and $\Omega$ values in Puget Sound surface waters during fall and winter months (November–March; Fassbender et al., 2018). Seasonal variability of $pCO_2$, pH, and $\Omega_{arag}$ observed in high-resolution moored surface time-series is among the highest in the world (Sutton et al., 2016), so these waters have a long "time of emergence" for detecting statistically significant anthropogenic trends in $CO_2$ content (Sutton et al., 2016, 2019). Moreover, biological modulation of carbonate chemistry or temperature seasonality can obscure or decouple changes in pH and $fCO_2$ from those seen in saturation states (Kwiatkowski and Orr, 2018; Lowe et al., 2019; Cai et al., 2020). Estimates of anthropogenic $CO_2$ content from Salish Sea observations and models point to widespread $\Omega_{arag}$ undersaturation having emerged here and in other regional waters since pre-industrial times (Feely et al., 2010; Pacella et al., 2018; Evans et al., 2019; Hare et al., 2020; Jarníková et al., 2022). These factors, in tandem with strong benthic-pelagic coupling of biogeochemical cycles (e.g., high surface productivity contributing to deep respiration hotspots, Hickey and Banas, 2008; Siedlecki et al., 2015), highlight the need for detailed biogeochemical observations throughout the water column in this biologically productive region to understand the atmospheric, terrestrial, and marine processes driving dynamic biogeochemical conditions in the Salish Sea.

Here we use the Salish cruise data product (2008–2018; Alin et al., 2022, 2023a, b) to characterize seasonal variability and major anomalies in physical and biogeochemical conditions in Puget Sound and its boundary waters (Strait of Juan de Fuca, coastal waters) during the seasonally resolved part of the time-series (2014–2018). All calculated marine inorganic carbon parameters used in this analysis were calculated from measured dissolved inorganic carbon, total alkalinity, and ancillary hydrographic observations (temperature, salinity, and phosphate and silicate content) described by Alin et al. (2023a). We used temperature, salinity, oxygen ($O_2$), fugacity of carbon dioxide ($fCO_2$), and aragonite saturation state ($\Omega_{arag}$) median conditions and variation to characterize seasonal ocean acidification, hypoxia, and warming conditions across Puget Sound basins and its boundary waters. Major anomalies in large-scale marine and atmospheric temperature, as well as regional precipitation and river runoff, occurred during 2013–2018, and we qualitatively relate the timing and magnitude of observed biogeochemical



anomalies in the study region to anomalies in regional weather and physical oceanography sometimes driven by these major large-scale anomalies. Cruises prior to the onset of the 2013–2018 anomalies and existing regional climatologies provided the

long-term context for the apparent magnitude and duration of physical and biogeochemical anomalies observed during the seasonal sampling period. Finally, we evaluated how the physical and biogeochemical Salish cruise time-series through 2018 reveals the changing landscape of multiple interacting ocean stressors as they are relevant to key ecologically and economically important fish and invertebrate species in this oceanographically dynamic region.

## 2 Environmental context for the Salish Sea cruise time-series

### 2.1 Geographic setting

The northern California Current Ecosystem (CCE) is the marine source water for deep waters within the southern Salish Sea and experiences episodic upwelling during spring–early fall (April–September) as a result of northwesterly equatorward winds causing offshore Ekman pumping (Huyer, 1983). Downwelling conditions occur during late fall–early spring (October–March) due to seasonal wind reversal to poleward-dominant winds along the coast. Upwelling conditions bring deep, nutrient- and

$CO_2$-rich, and $O_2$-depleted marine water masses into the Strait of Juan de Fuca (SJdF) from the Juan de Fuca Canyon. This water transits at depth to the glacial sill complex at Admiralty Reach (AR), where it enters Puget Sound at depth during episodic marine intrusions. Strong freshwater outflow through SJdF, particularly during summer months when peak Fraser River discharge occurs, co-drives this estuarine circulation (Davis et al., 2014; Giddings et al., 2014). The glacial sills within AR at the entrance to Puget Sound impart strong mixing—of outgoing warmer, fresher surface estuarine waters with colder, saltier

marine waters entering Puget Sound from SJdF at depth—changing the characteristics of the incoming marine water that refreshes deep water masses in all PS basins and refluxing back some of the outgoing estuarine water.

Puget Sound (PS) is the southernmost glacial fjord estuarine system on the North American Pacific Coast and the southernmost part of the Salish Sea, which also encompasses the Strait of Juan de Fuca and the Strait of Georgia (Figure 1). Puget Sound is

comprised of four basins: Main Basin, South Sound, Whidbey Basin, and Hood Canal. The region directly receives freshwater input from 12 major and many smaller rivers draining into PS and the indirect input of the Fraser River, which drains into the Strait of Georgia, in addition to carbon and nutrient inputs from urban and agricultural environments surrounding the Salish Sea ecosystem (Mohamedali et al., 2011; Banas et al., 2015). Glacial sills restrict estuarine circulation throughout the Salish Sea and among Puget Sound basins, limiting marine intrusions and deep-water renewal to episodic occurrences and resulting

in long residence and flushing times in some parts of this inland sea, including Hood Canal (Babson et al., 2006; Pawlowicz et al., 2007; MacCready et al., 2021). The Main Basin (MB) is the widest, deepest, and most deeply wind-mixed of the Puget Sound basins. Deep waters enter South Sound, the shallowest basin, from MB when they pass over another glacial sill and undergo strong tidal mixing again while passing through the Tacoma Narrows. Thus, the deeply mixed Main Basin (MB) and South Sound (SS) share a deep-water transit path. In contrast, Whidbey Basin (WB) and Hood Canal (HC) have narrower





basins than MB and major river inputs emanating from the terminus of each basin, resulting in strong stratification and gradients of physical and biogeochemical conditions between surface and bottom waters. HC is also bounded by a glacial sill and has a long history of study of deep-water oxygen concentrations, as hypoxia and fish kills have been observed there (Newton et al., 2011, 2012 and references therein). Notably observations and models for the Strait of Georgia suggest that mixing associated with glacial bathymetric features to the north of Puget Sound may afford some protection to deep northern

Salish Sea basins, due to more rapid $O_2$ uptake than $CO_2$ outgassing (Johannessen et al., 2014; Ianson et al., 2016); this mechanism does not appear to protect Hood Canal from developing hypoxia.  While not bounded by a glacial sill, circulation in WB is severely restricted at its northern outlet, and it receives strong river input in two locations. While WB has side inlets with hypoxia, the mainstem of the basin tends to see only moderately low oxygen values but not hypoxia.

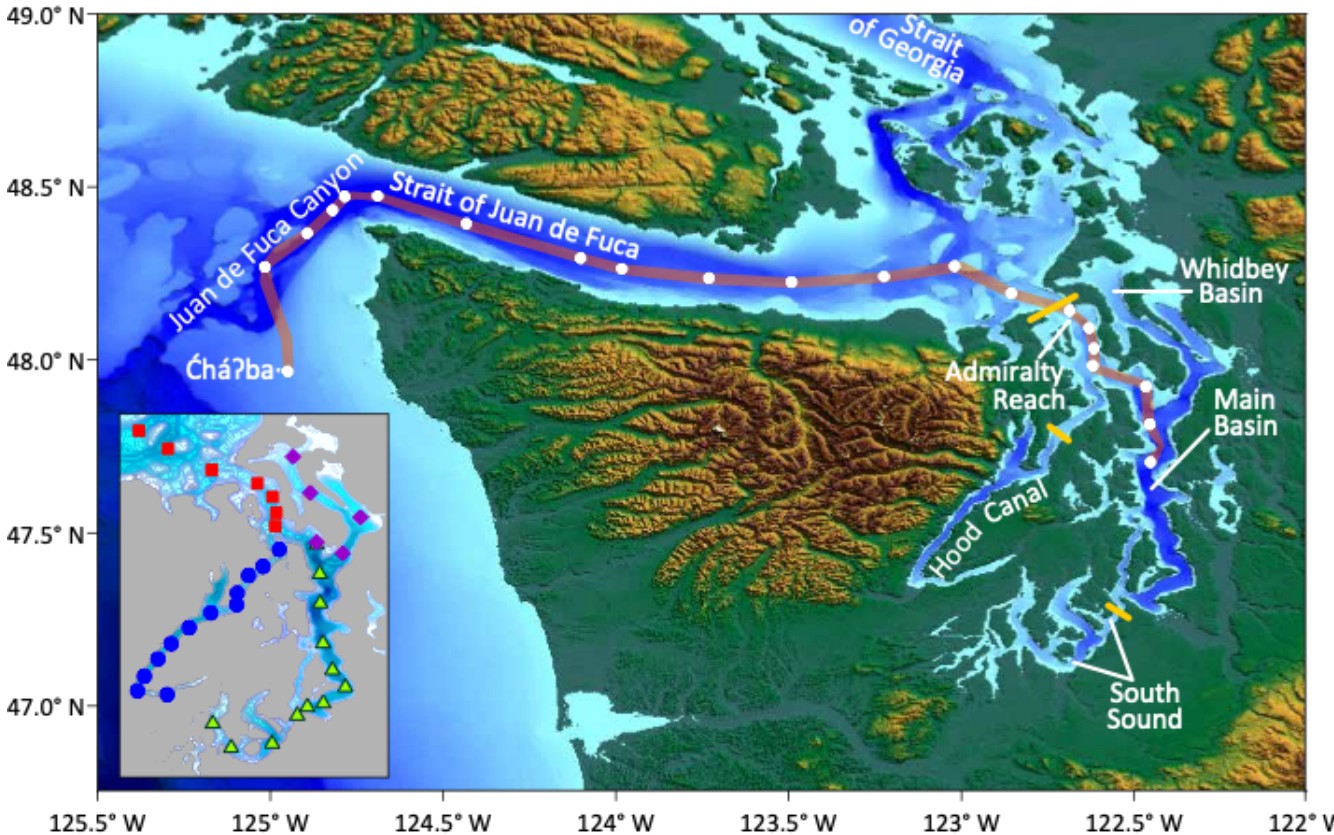

Figure 1: Map of the southern Salish Sea and its boundary waters with all study basins named. The subset of sampling stations tracing a path between the Ćháʔba· mooring on the Washington state (USA) continental shelf to the Main Basin of Puget Sound constitute the Sound-to-Sea (S2S) transects.  Inset map shows station groupings used for analyses of Puget Sound cruises: Admiralty Reach (AR, red squares), Main Basin–South Sound (MB–SS, green triangles), Whidbey Basin (WB, purple diamonds), and Hood

Canal (HC, blue circles). Yellow dashes denote locations of glacial sills that restrict deep-water exchange. The Fraser River, mentioned in the text, enters the Strait of Georgia from the east, to the north of the map area. Topographic and bathymetric data were extracted from the NOAA National Centers for Environmental Grid Extract Coastal Relief Model (3-second resolution,





**https://www.ncei.noaa.gov/maps/grid-extract/, accessed Nov. 13, 2014), and data were gridded in Surfer using a minimum curve gridding technique.**

Both regional weather and large-scale climate factors play important roles in driving physical, chemical, and biological processes in the Salish Sea and its boundary waters. From 2013 to 2016, an unprecedented marine heatwave (MHW) developed and persisted in the NE Pacific Ocean, followed by a very strong El Niño event in the equatorial Pacific Ocean during 2015–2016, both of which strongly influenced regional weather, oceanography, and ecosystems (e.g., Bond et al., 2015; Jacox et al., 2016; McClatchie et al., 2016; Morgan et al., 2019; N. Bond in Sobocinski, 2021). The NE Pacific heatwave's direct influence

on Washington's coastal waters and the Salish Sea ecosystem began when anomalously warm waters from the North Pacific were advected onto the Pacific Northwest coast in mid-September 2014 (Peterson et al., 2017). However, associated strong, large-scale air temperature anomalies strongly influenced the surface Puget Sound system and preceded the arrival of the warmed ocean water masses (Swain et al., 2016), with anomalously warm, dry summer conditions starting in 2013 over the southern Salish Sea (Table 1, Figure 2). As a result of these large-scale heat anomalies, Puget Sound and Washington coastal

waters also experienced strong precipitation, river discharge, and solar energy flux anomalies during 2013–2018 (Table 1 and references therein). Upwelling anomalies reflect basin-scale climate drivers and influence upwelling strength and depth of source waters for deep waters of the southern Salish Sea (e.g., Jacox et al., 2015).

**Table 1.** Major environmental anomalies occurring during 2013–2018 and regional environmental drivers affecting the
southern Salish Sea.

| | weather anomalies | | | hydrological anomalies | circulation driver anomalies and effects | | |
| | | | | | | | |
| Year | Air temperature (°C)[a] | Solar energy flux[b] | Precipitation[a] | River discharge (Q)[c] | Upwelling[d] | Puget Sound circulation and stratification[e] | Sources[f] |
|---|---|---|---|---|---|---|---|
| 2013 | +0.5–0.9 May–Sep. | Higher than normal May–Aug. | Apr. and Sep. very wet, Oct.–Dec. drier than normal | FR: high early peak Q in May, low fall Q; PS: higher Q in Mar.–Jun. and early Oct. | Below normal Aug.–Sep. | Stratification started earlier; normal marine intrusion timing (fall) | Bumbaco[a]; Albertson et al.[b,d]; Dzinbal[c]; Ruef et al.[e] |
| 2014 | 0.9+ (5th warmest year on record) | Near theoretical maximum May–Sep. | 119% (wettest Mar.) | FR: high early peak Q in May, low summer Q, high late fall Qs; PS: higher than normal in spring and fall, average in summer | Early fall transition to downwelling (stronger than normal in Sep.–Oct.) | Deep mixing driven by cold, dry, windy conditions led to persistent (Feb.–Oct.) high O$_2$, low T conditions in deep HC; stronger than normal stratification in parts of MB | Bumbaco[a]; Albertson et al.[b,d]; Dzinbal[c]; Mickett et al.[e]; Stark[e] |
| 2015 | 1.4+ (warmest year on record) | Near theoretical maximum May–Sep.; gloomy fall | 107%, snowpack deficit, summer drought (3rd wettest Dec.) | FR: record high spring Qs, high/early peak, record low summer Qs; PS: extremely high Jan.–Mar., Nov.–Dec.; extremely low May–Sep./Oct. | Stronger May–Jun. upwelling | Strong early stratification in MB, reduced by summer drought; HC deep-water renewal six weeks early (deep water >2.5 °C above climatology) | Bumbaco[a]; Albertson et al.[b,d]; Dzinbal[c]; Bos et al.[e]; Ruef et al.[e] |
| 2016 | 1.0+ (3rd warmest | Below average winter and fall; above | 113% (wet Feb.–May, normal | FR: early low peak (4–6 weeks early), very low summer, very high Nov.; | Stronger downwelling Jan.–Mar., | Stronger stratification than normal in spring and fall; longer residence time in | Bond and Bumbaco[a]; Albertson et |



| Year | Air temp | Solar | Precip | Rivers | Upwelling | Marine | Sources |
|---|---|---|---|---|---|---|---|
| year on record) | | average spring–summer | summer, wettest Oct.) | PS: very high Qs mid-Jan.–Mar., Oct.–Nov., low May–Sep. Qs | Oct–Nov.; stronger May upwelling | MB during summer drought; annual flushing of deep HC water four weeks early | al.[b,d]; Burks[c]; Bos et al.[e]; Ruef et al.[e]; Albertson et al.[e] |
| 2017 | Normal (warmest August) | Below average Feb.–May, Nov.; above average Jun.–Oct. (Jul.–Sep. periods of haze to wildfire smoke) | 112% (wettest Feb.–Apr., driest Jul.–Sep., wet fall, normal snowpack) | FR: early higher peak Q, lower Q Jul.–Oct., high Nov.–Dec.; PS: high peaks in Jan.–Mar., low end of normal Qs through summer, event peaks in Oct., Nov. | Stronger downwelling in Apr., Nov. | Stronger, more persistent stratification than normal due to high spring Q anomaly (resulting in sustained MB phytoplankton blooms, persistent low water column S in MB); normal HC deep flushing, historic salinity minima (2 SD below normal before summer) | Bumbaco and Bond[a]; Albertson et al.[b,d,e]; Burks[c]; Bos et al.[e]; Ruef et al.[e] |
| 2018 | 0.6+ | Above average except Jan., Feb., Apr. (Aug.–Sep. reduced due to wildfire smoke) | 98% (2nd wettest Apr., driest May–Aug.) | FR: runoff peak one month early, highest peak of these years, low Qs Jun.–Nov.; PS: very high Q peaks in winter–spring (early) and fall, low-to-very-low summer Qs | Stronger downwelling in Jan.; upwelling in Feb. (spring transition two months early) | Strong springtime density stratification, but decreased in summer (delayed bloom); more favourable periods for intrusions across AR than any year since 2013; normal timing for deep HC renewal (by end Sep.) | Bumbaco and Bond[a]; Albertson et al.[b,d,e]; Burks[c]; Bos et al.[e]; Szuts et al.[e] |

[a] Annual and monthly average air temperature and precipitation anomalies are relative to 1981–2010 normal. Record monthly, seasonal, or annual anomalies up to 2018 noted in parentheses. [b] Daily solar energy flux, an indicator of sunniness, measured in Seattle and compared to the highest theoretical solar energy for the latitude and time of year and fully overcast conditions. [c] River flows for the Fraser River (FR)—with a single summer discharge (Q, m$^3$ s$^{-1}$) peak—are described relative to the median discharge from 1912 to the year reported. Puget Sound rivers (PS)—with an early summer snowmelt discharge peak, and a late fall rainfall and winter storm discharge peak—are reported relative to the full length of their U.S. Geological Survey observational records, ranging from 47 to 104 years. [d] Upwelling anomalies are reported as monthly average upwelling index values (m$^3$ s$^{-1}$ per 100 m of coastline) that fall outside the interquartile range (25–75%) for 48° N, 125° W by NOAA's Pacific Fisheries Environmental Laboratory. Baseline period is 1967 to the year of each annual report. [e] Anomalies in Puget Sound stratification or deep-water renewal events were reported in temperature and salinity water quality narratives of annual marine conditions reports. [f] Sources for observations in previous columns are listed by authors of the relevant sections in the PSEMP Marine Waters Workgroup annual overview of marine conditions published in the following year (i.e., listed as citations in the relevant year's report: PSEMP Marine Waters Workgroup, 2014, 2015, 2016, 2017, 2018, 2019).

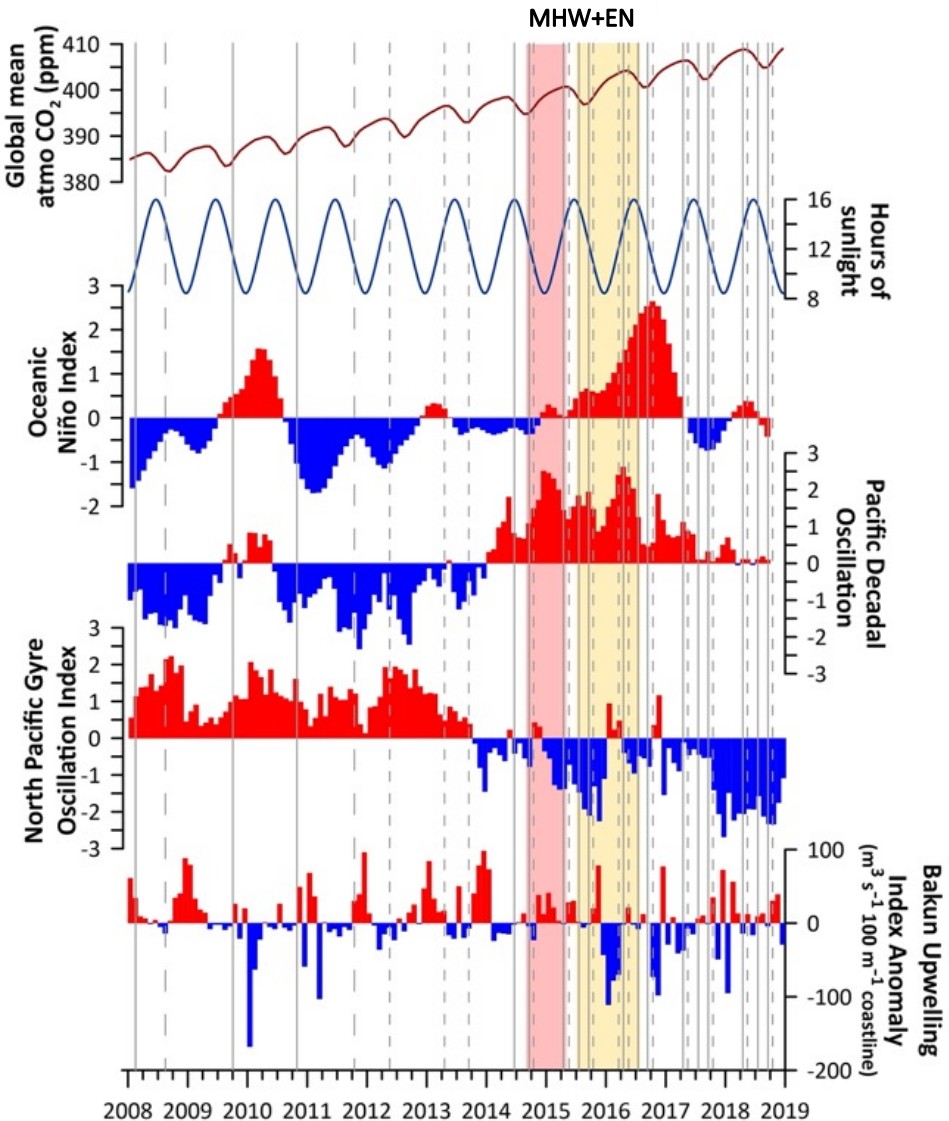

**Figure 2:** Monthly time-series for 2008–2018 for the Oceanic Niño Index anomaly (NOAA Climate Prediction Center, 2019), Pacific Decadal Oscillation (Mantua, N., 2019), North Pacific Gyre Oscillation (Di Lorenzo, 2019), and Bakun Upwelling Index Anomaly for 48° N (NOAA Pacific Fisheries Environmental Laboratory, 2019). Positive anomalies for all climate indices are shown in red, negative in blue. Superimposed on this are the durations of the maximum intensity of the northeast Pacific marine heatwave (MHW), shaded in red during its peak manifestation in Washington marine waters (September 2014–April 2015) and in yellow for its later moderate intensity window, which overlapped with the 2015–2016 El Niño event (EN, July 2015–July 2016). Duration and intensity ranges were inferred from Gentemann et al. (2017) and the OSU MODIS water temperature anomaly climatology tool (NANOOS, 2019). Overlap between the 2014–2016 NE Pacific heatwave and warm waters that may be the result of the El Niño can be seen by comparing the Oceanic Niño Index positive anomalies to the yellow shading, based on satellite sea surface temperature analysis associated with the marine heatwave (i.e., Gentemann et al., 2017). Seasonality is shown using hours of sunlight per day as a proxy, shown at top (timeanddate.com, 2019). Finally, the timing of all Salish cruises is indicated by vertical lines, with Sound-to-Sea (S2S) cruises in short-dashed lines, Puget Sound (PS) cruises in solid lines, and the two cruises that encompassed both sets of stations in long-dashed lines.



## 3 Methods: Observations, calculations, and data visualization

### 3.1 Salish cruise time-series

For this analysis of seasonal variability and oceanographic anomalies, we used the Salish cruise data product, comprising 35 consistently formatted and quality controlled cruises data sets collected throughout the study region from 2008 to 2018 (Alin et al., 2022, 2023a).  The timing of both Puget Sound (PS) cruises and Sound-to-Sea (S2S) cruises has been particularly consistent and frequent since 2014, with 24 of the 35 cruises having taken place between July 2014 and October 2018. Thus, our analysis of seasonal patterns in ocean conditions spanning the upwelling season focuses on the period from 2014 to 2018
(Figure 2), and will provide oceanographic context for numerous biological oceanography studies that have also been conducted on the S2S and PS cruises. While each cruise represents a snapshot of conditions along a transect from the coast into the southern Salish Sea or across the basins within the southern Salish Sea, collectively this cruise time-series illuminates typical spatial patterns throughout the southern Salish Sea, throughout the water column, and through the seasonal cycle.

### 3.2 Calculated parameters and uncertainty

DIC and TA measurements from the Salish cruise data product were used to calculate the full suite of inorganic carbon parameters, although our discussion of seasonal ocean acidification conditions and anomalies focuses on the calculated parameters $f\mathrm{CO_2}$ and $\Omega_{arag}$, as these are two of the inorganic carbon parameters most familiar to our science and resource management end users. Of the carbonate system parameters, $f\mathrm{CO_2}$ is most directly relatable to atmospheric values and trends and thus has intuitive value, and $\Omega_{arag}$ is critical to many of the important calcifying species in the Salish Sea. For species and
investigators for whom $pH_T$ and $\Omega_{calc}$ are more relevant (e.g., Dungeness crab), complementary figures are provided in the Supplemental Material. A regression between $\Omega_{arag}$ and $\Omega_{calc}$ had a coefficient of determination ($R^2$) >0.99, with $\Omega_{calc}$ being 1.59 times $\Omega_{arag}$ in the Salish cruise data package, which is a slightly larger coefficient than reported for the relationship between the two parameters by Mucci (1983), presumably to different temperature and salinity characteristics of our regional water masses.


We used the *R* package *seacarb* function *carb* to calculate all carbonate system parameters (Gattuso et al., 2023). Input parameters from the Salish cruise compiled data set comprised DIC (DIC_UMOL_KG), TA (TA_UMOL_KG), phosphate (PHOSPHATE_UMOL_KG), and silicate (SILICATE_UMOL_KG) content values from bottle samples analyzed in the laboratory, along with CTD measurements in the field of temperature (CTDTMP_DEG_C_ITS90), salinity
(CTDSAL_PSS78), and pressure (CTDPRS_DBAR). Within *seacarb*, we used the TEOS-10 thermodynamic seawater equations option (IOC, SCOR, and IAPSO, 2010). For equilibrium constants ($K_1$ and $K_2$), we did two sets of calculations, using Lueker et al. (2000, "L00") for one and Waters et al. (2014, "W14") for the other, with all other *seacarb* options the same. We adopted the total scale for pH ($pH_T$), the Uppstrom (1974) formulation for deriving total boron concentration from salinity, the *seacarb* default option for $K_f$ (Perez and Fraga, 1987 for temperatures above 9 °C; Dickson and Goyet, 1994 for



those below), and the Dickson (1990) option for $K_s$ (following results of Orr et al., 2015). All input content data were first divided by $10^6$ to convert from µmol kg$^{-1}$ to mol kg$^{-1}$, and pressure (dbar) was divided by 10 to convert to bar, to conform with the default units of *seacarb*. Calculated values of $f$CO$_2$ shown in figures here are the *seacarb* "*in situ*" CO$_2$ fugacity values, referenced to *in situ* temperature and pressure, rather than atmospheric pressure as the "standard" and "potential" options are computed (Gattuso et al., 2023), because *in situ* results are more germane to understanding the environmental conditions

confronting populations of marine organisms in the wild. Total uncertainties on calculated values using high-quality DIC and TA measurements as input parameters are ±3.5% for $f$CO$_2$ (±14 µatm at 400 µatm and ±70 µatm at 2000 µatm) and ±4.9% for $\Omega_{arag}$ (±0.025 at $\Omega_{arag}$ = 0.5, ±0.049 at $\Omega_{arag}$ = 1, and ± 0.075 at $\Omega_{arag}$ = 1.5) (per Orr et al., 2018).

In order to facilitate comparison with results from West Coast Ocean Acidification (WCOA) cruise publications (e.g., Feely

et al., 2008, 2016), we used in L00 results in figures, statistics, and discussion. However, for end users using dissociation constants explicitly targeted to the broader salinity ranges seen in the Salish Sea (i.e., W14 constants) for their own measurements, we did a comparison between calculated values using L00 and W14 dissociation constants. This comparison showed that the average differences in calculated values using L00 rather than W14 constants (i.e., L00 – W14) were – 0.003±0.002 for pH$_T$, +5.0±1.3 µatm for $f$CO$_2$ (+5.1±1.4 for $p$CO$_2$), and –0.001 for $\Omega_{arag}$ and $\Omega_{calc}$ (±0.003 and ±0.004,

respectively). Only two samples (of 4021) of those with good analytical values for all analytical input parameters had salinity <19—the lower end of the salinity range used by Lueker et al. (2000), and thus generated warning flags. Thus, our use of the L00 constants would not have a discernible effect on the results discussed here. For the L00 vs. W14 comparison, we used the TEOS-10 thermodynamic seawater equations with both sets of carbonate system dissociation constants. To facilitate use of this data set, including the calculated values discussed here, we created a new multi-stressor Salish cruise data product that

includes all of the highest quality *seacarb* input data (i.e., temperature, salinity, and DIC, TA, phosphate, and silicate content with quality flags of 2, indicating "acceptable" data quality), along with high-quality recommended O$_2$ values, and the most commonly used CO$_2$ system parameters (pH$_T$, $f$CO$_2$, $p$CO$_2$, $\Omega_{arag}$, and $\Omega_{calc}$) calculated using both L00 and W14 dissociation constants (Alin et al., 2023b).

However, prior WCOA cruise publications used EOS-80 seawater equations, whereas we used the more recent TEOS-10 equations, in keeping with current recommendations (IOC, SCOR, and IAPSO, 2010; Jiang et al., 2022). The differences between paired calculated carbonate system parameters caused by using EOS-80 vs. TEOS-10 thermodynamic seawater equations with the L00 dissociation constants were similar in magnitude, with average offsets (i.e., EOS-80 – TEOS-10) of – 0.003 for pH$_T$, +5.2 µatm for $f$CO$_2$, 0.000 for $\Omega_{arag}$, and –0.001 for $\Omega_{calc}$.


Finally, we use $f$CO$_2$ rather than $p$CO$_2$ as it provides the most accurate estimate for *in situ* gas-phase CO$_2$, because it accounts for interactions between CO$_2$ and other molecules in seawater (per recommendations in Jiang et al., 2022). To facilitate comparisons between $f$CO$_2$ values and $p$CO$_2$ values from other sources, the average difference between the two values (i.e.,



$p$CO$_2$ – $f$CO$_2$ because $p$CO$_2$ is always larger) in this data compilation is 22.5 µatm across all pairs of seawater and carbonate
system thermodynamic constants. The difference increased with $f$CO$_2$ level, with $f$CO$_2$ values averaging 2.8 µatm lower than
$p$CO$_2$ at 0–499 µatm, 17.7 µatm at 500–999 µatm, 43.4 µatm at 1000–1999 µatm, and 55.7 µatm at >2000 µatm.

### 3.4 Data visualization

Raincloud plots display both raw data and percentile distributions to provide transparent statistical data summaries (Allen et
al., 2021). We use them to visually summarize the 2014–2018 statistical distributions of temperature, salinity, oxygen, $f$CO$_2$,
and Ω$_{arag}$ observations from May and October boundary water (S2S) cruises in Figures 3A–4A and 6A–8A and for April, July,
and September PS cruises in Figures 3B–4B and 6B–8B. Raincloud statistical summaries for potential density anomaly (sigma
theta, $\sigma_\theta$), Ω$_{calc}$, and pH$_T$ are provided in Figures S1–S3 for readers interested in this information. Raincloud plots were created
using *R* code by Cédric Scherer (Scherer, 2021) but modified extensively for use with Salish cruise data. To characterize
differences in median and extreme values for all parameters with depth, we used 20 dbar as the boundary between surface and
subsurface depth categories throughout the region, although we acknowledge that mixing depths vary across the study region
and seasons, such that the upper mixed layers occupy different depth ranges through space and time.

Bubble cloud plots are scatterplots where additional statistics can be represented by the size and colour of each data point. We
used the *R ggplot2* package *ggpubr* functions *facet_grid* and *geom_point* to create bubble plots summarizing seasonal changes
in median values and ranges of T, S, O$_2$, $f$CO$_2$, and Ω$_{arag}$ across the study region and by depth during 2014–2018 (Figure 5).

### 4 Results: Seasonal variability of physical and biogeochemical parameters across depths and basins during 2014–2018

In this section, we describe seasonal oceanographic variation within and across basins for the latter half of the time-series
(2014–2018), including differences between surface and subsurface water masses, with ranges serving as our metric of
variability. We note the timing and magnitude of apparent anomalies in median or variability for each parameter here and
relate these physical and biogeochemical Salish cruise anomalies to the major 2013–2018 weather, hydrological, and
circulation anomalies (Table 1) in Section 5, using existing climatologies to provide longer-term context. We refer to cruises
occurring in April and May as "spring" cruises, July cruises as "summer," and September–October cruises as "fall." To denote
specific cruises, we abbreviate the cruise by the first letter of the month (A, M, J, S, and O, respectively) and the two-digit year
(e.g., October 2014 becomes O14). "Spring" cruises also reflect early upwelling season conditions, while July–September
cruises represent late upwelling season conditions. "Coastal" refers to stations outside the mouth of the Strait of Juan de Fuca
(SJdF), sampling either deep Juan de Fuca Canyon (JdFC) stations or the Ćháʔba· station in shallower water on the continental
shelf (Figure 1). All observations in this compiled cruise data product reflect open basin conditions throughout Puget Sound
and its boundary waters, which may be quite different from nearshore environments, such as the finger inlets of South Sound
or seagrass meadows that may have markedly different circulation, freshwater influence, and retention times.





### 4.1 Physical oceanographic seasonality across basins

During 2014–2018, coastal surface temperature and salinity are largely dominated by seasonal upwelling/downwelling dynamics, as expected. Coastal surface temperatures spanned similar ranges in the early and late upwelling season, whereas deep coastal water had more of a seasonal contrast between early and late upwelling season, with warmer water and wider T ranges in the fall (Figure 3A, Table 2). Both surface and subsurface temperatures were warmer by nearly 3 °C during O14, when the MHW was strongest in coastal waters, with less elevated temperatures in surface waters in M16 during the El Niño. Surface salinities were lower in spring than fall, with somewhat fresher anomalies during M17 and O15. October cruises during 2016–2018 had higher surface salinities than O14 and O15 (Figure 4A). However, deep fall salinities during 2014–2015 occupied larger ranges (Table 2). Potential density anomaly values (sigma theta, $\sigma_\theta$) track salinity quite closely throughout this region (Moore et al., 2008) and are not discussed further here but are represented in Figure S1. Depth distributions of physical parameters can be seen in more detail in Figures 4–6 in Alin et al. (2023a).

Seasonal patterns in Strait of Juan de Fuca surface waters were similar, with somewhat narrower surface temperature and salinity ranges than at coastal stations across the upwelling season (Figures 3A, 4A). As seen at coastal stations, subsurface temperatures and salinities in SJdF showed stronger variability, particularly in fall (Table 2). Temperature anomalies manifested as the widest ranges and highest medians during O14 across the water column, with some residual heat persisting in the form of wider ranges and higher medians across the water column in O15 and O16 relative to O17 and O18 when medians were lower. Subsurface salinity had higher medians than at the surface on all cruises and occupied wider ranges except during M18. Salinity had lower median values across the water column during O14, O15, and O16 than O17 and O18.

AR showed a seasonal progression between April and July–September cruises of median temperatures warming by 1–2° C and variability increasing across depths (Figure 3B). April temperatures were warmer across depths during 2015–2016 relative to 2017–2018, and a similar magnitude of warming was observed across depths between J14 and J15 (Table 2). The seasonal span of salinities at AR overlapped across depths, with greater variability at depth in all seasons (Figure 4B). Median salinity increased across depths by a few salinity units from April to September each year as upwelled deep coastal waters arrive at AR by the end of the upwelling season. Median 2017 AR salinities decreased by ~1 across the water column compared to other years, except at depth in S17.





**Figure 3: A)** Raincloud plots for CTD temperature in coastal (upper row) and Strait of Juan de Fuca (lower row) surveys in the early and
late upwelling season beginning in the fall of 2014. Cruise timing is indicated with a one letter month (M=May, O=October) and a two-digit
year. Surface observations are in the left column, and subsurface observations are in the right column. Percentiles for observations are





reflected by the colours of the vertical bars, similarly to a box plot, with the median displayed to the right of each bar as an unfilled black diamond and individual observations plotted to the left of each vertical bar as transparent grey circles. Note that all panels in this figure have the same scale bar, but differ from those in the corresponding Puget Sound figure. **B)** Raincloud plots for CTD temperature in Puget Sound

regions—Admiralty (top row), Main Basin (second row), South Sound (third row), Whidbey Basin (fourth row), and Hood Canal (bottom row)—in April, July, and September beginning in July 2014. Cruise timing is indicated with a one letter month (A=April, J=July, S=September) and a two-digit year. Surface observations are in the left column, and subsurface observations are in the right column. Percentiles for observations are reflected by the colours of the vertical bars, similarly to a box plot, with the median displayed to the right of each bar as an unfilled black diamond and individual observations plotted to the left of each vertical bars as transparent grey circles.








**Figure 4: A)** Raincloud plots for CTD salinity in coastal and Strait of Juan de Fuca surveys in the early and late upwelling season beginning in the fall of 2014. Figure organization is the same as in Fig. 3A. **B)** Raincloud plots for salinity in Puget Sound surveys in April, July, and September beginning in July 2014. Figure organization is the same as in Fig. 3B.

MB and SS are the least strongly stratified basins within Puget Sound; they show relatively weak gradients and narrow ranges in temperature and salinity from surface to deep waters, despite MB bottom depths reaching ~225 dbar (Figures 3B, 4B, Table 2). Due to deep mixing, MB and SS are often the warmest at depth of all basins. Temperatures increased by 2–4 °C between April and July–September cruises in both surface and deep water, with typically 1–3 °C difference between surface and deep median temperatures (Figure 3B). Salinity across depths showed progressive increases of ~1–3 across April–September cruises in a given year in both basins, with low variability across the water column (difference of ~1 or less between surface and deep

medians, Figure 4B). April temperatures were ~2 °C higher across depth in both basins during 2015–2016 than 2017–2018. Median surface temperatures in J15 and J18 were elevated by ~1–2 °C across MB–SS, and deep J15 and O15 median temperatures were <1 °C higher than in other years (Figure 3B). High salinity outliers (by >3 salinity units) were seen across depth in both basins during A15, and relatively low surface salinity medians were observed during A17 (outliers to ~24) and J17 in MB and SS (Figure 4B, Table 4).


WB and HC have the strongest stratification and gradients of physical and biogeochemical conditions between surface and bottom waters in PS (Figures S5–S8 here and Figures 5–9 and S1–S4 in Alin et al., 2023a). At the entrance to HC, deep-water replacement is constrained by an additional glacial sill and influenced by further mixing of surface with deep water. Subsurface waters in both basins warmed continuously from April through September cruises each year, as they did in MB (Table 2).

Deep-water salinity also increased steadily between April and September across WB and HC, with the only obvious anomaly among years being lower salinity in deep WB waters during A17 (lower by ~2). Surface waters in WB and HC were the most variable of all regions for temperature and salinity. The widest ranges of surface temperature occurred in WB and HC during Julys, though median surface temperatures were similar between Julys and Septembers of each year in WB and warmest during July cruises in HC, with cooling by Septembers. Surface salinity tended to be highest and ranges narrower in Septembers, with

considerable interannual variability during 2014–2018. Surface temperature anomalies were seen in higher medians in J15 and J18 in HC and J15 in WB, and at depth in both basins in S15, while A15 and A16 had medians ~2 °C warmer across depth in both basins than A17 and A18 (Figure 3B). A16 and A17 had lower median salinities in both WB and HC than A15 or A18 (Figure 4B). All 2017 cruises had anomalously wide ranges and low outliers for salinity in HC.

Puget Sound temperature ranges were mostly shifted up relative to the boundary waters across depths, with higher medians and wider ranges everywhere in summer and fall than spring (Figure 5, Table 2). As expected, median salinity was consistently fresher in PS than boundary waters, with substantially wider overall ranges. Vertical temperature and salinity gradients (surface median – deep median) were weakest at AR and SS stations. Temperature gradients were weakest in spring and strongest in summer, while salinity gradients were strongest in spring and weakest in fall. HC summertime temperature gradients were the



strongest (4.5° C), with WB usually having the strongest salinity variability and surface–deep gradients (4.5 salinity units). While surface variability in T and S tended to be higher in PS surface waters, boundary waters typically showed greater subsurface variability.



**Figure 5:** Bubble plots of summary statistics for all five parameters—temperature, salinity, oxygen content, fugacity of $CO_2$ ($fCO_2$), and aragonite saturation state ($\Omega_{arag}$) in surface (left column) and deep (right column) water. Bubbles are plotted by the magnitude of mean monthly medians for each parameter taken across Washington Ocean Acidification Center (WOAC, April, July, and September) and Sound-to-Sea (S2S, May and October) cruises during 2014–2018. Bubbles are filled with colours representing the basin the observations were





derived from (CO=Coast, SJdF=Strait of Juan de Fuca, AR=Admiralty Reach, MB=Main Basin, SS=South Sound, WB=Whidbey Basin, HC=Hood Canal). The area of the bubble represents the average range width for that parameter across 2014–2018 WOAC or S2S cruises.
The bubble sizes in the legend represent the upper ends of four bins of average range widths (i.e., maximum – minimum) for each parameter: temperature range bins are 0.0–2.5 °C, 2.5–5.0 °C, 5.0–7.5 °C, and 7.5–10.0 °C; salinity range bins are 0.0–2, 2–4, 4–6, and 6–8; oxygen content bins are 0.0–62.5 µmol kg⁻¹, 62.5–125 µmol kg⁻¹, 125.0–187.5 µmol kg⁻¹, and 187.5–250.0 µmol kg⁻¹; $fCO_2$ bins are 0–600 µatm, 600–1200 µatm, 1200–1800 µatm, and 1800–2400 µatm; and $\Omega_{arag}$ bins are 0.1–0.6, 0.6–1.1, 1.1–1.6, and 1.6–2.1. Bubbles are ordered such that those with the largest ranges are at the back. Thus, if a basin is not visible, its range overlaps completely with another basin's range.


**Table 2.** Ranges of surface (≤ 20 dbar) and deep (>20 dbar) temperature (T, ITS-90), salinity (S, PSS-78), oxygen ($O_2$), and calculated values of $CO_2$ fugacity ($fCO_2$) and aragonite saturation state ($\Omega_{arag}$) for all regions.

| Region & month | Surface T (° C) | Deep T (° C) | Surface S | Deep S | Surface O₂ (µmol kg⁻¹) | Deep O₂ (µmol kg⁻¹) | Surface fCO₂ (µatm) | Deep fCO₂ (µatm) | Surface Ω_arag | Deep Ω_arag |
|---|---|---|---|---|---|---|---|---|---|---|
| *Coast* | | | | | | | | | | |
| May | | | | | | | | | | |
| 2015 | 9.3–11.2 | 6.2–9.9 | 31.3–32.1 | 32.1–34.0 | 206–387 | 77–250 | 221–713 | 635–1183 | 1.07–2.70 | 0.66–1.16 |
| 2016 | 11.3–13.1 | 7.6–10.2 | 31.3–31.3 | 31.9–33.8 | 319–391 | 129–220 | 189–364 | 507–888 | 1.86–3.19 | 0.89–1.43 |
| 2017 | 10.1–10.9 | 8.2–9.9 | 30.2–30.7 | 32.0–33.7 | 311–332 | 133–293 | 281–289 | 306–855 | 2.06–2.18 | 0.94–2.20 |
| 2018 | 9.8–11.7 | 6.7–8.8 | 31.3–32.1 | 32.2–34.0 | 280–408 | 82–176 | 171–525 | 360–1177 | 1.37–3.25 | 0.67–1.82 |
| October | | | | | | | | | | |
| 2014 | 14.6–15.6 | 7.6–15.6 | 30.7–31.9 | 30.8–33.8 | 241–250 | 78–248 | 365–432 | 371–1399 | 1.83–2.27 | 0.58–2.18 |
| 2015 | 11.9–12.4 | 7.8–12.7 | 29.9–31.6 | 31.3–33.8 | 254–275 | 106–261 | 340–378 | 358–1005 | 1.87–2.02 | 0.79–2.04 |
| 2016 | 12.7–12.7 | 8.3–12.7 | 32.3–32.3 | 32.3–33.6 | 265–267 | 118–268 | 379–396 | 376–991 | 1.95–2.02 | 0.82–2.03 |
| 2017 | 9.3–11.2 | 7.6–9.2 | 32.1–32.5 | 32.5–33.9 | 141–268 | 52[b]–171 | 754–1042 | 668–2376 | 0.87–1.16 | 0.38–1.19 |
| 2018 | 11.6–12.5 | 8.0–10.4 | 32.3–32.4 | 32.6–33.9 | 281–326 | 82–257 | 300–359 | 321–1101 | 2.04–2.40 | 0.75–1.96 |
| | | | | | | | | | | |
| *Strait of Juan de Fuca* | | | | | | | | | | |
| May | | | | | | | | | | |
| 2015 | 9.1–10.1 | 6.8–9.9 | 30.5–32.0 | 30.7–33.9 | 193–221 | 96–220 | 636–777 | 669–1058 | 0.96–1.11 | 0.74–1.07 |
| 2017 | 9.2–9.9 | 7.8–9.0 | 30.3–31.2 | 31.2–33.8 | 262–429 | 120–245 | 157–528 | 559–942 | 1.23–3.09 | 0.85–1.27 |
| 2018 | 9.6–10.7 | 7.1–8.3 | 30.6–32.1 | 32.9–33.9 | 222–357 | 90–144 | 310–993 | 712–1115 | 0.78–2.15 | 0.72–1.09 |
| October | | | | | | | | | | |
| 2014 | 11.4–14.7 | 7.8–15.2 | 30.3–31.3 | 30.5–33.6 | 201–248 | 90–245 | 394–840 | 405–1121 | 0.96–1.94 | 0.66–1.92 |
| 2015 | 10.0–11.8 | 8.1–11.5 | 30.2–31.2 | 31.0–33.6 | 230–260 | 113–239 | 436–751 | 512–1044 | 0.97–1.55 | 0.77–1.38 |
| 2016 | 10.1–11.7 | 7.8–11.3 | 30.7–31.6 | 31.1–33.7 | 186–257 | 80–235 | 432–878 | 578–1269 | 0.89–1.67 | 0.65–1.30 |
| 2017 | 9.6–10.2 | 7.7–9.7 | 31.4–31.9 | 31.7–33.9 | 168–195 | 89–158 | 654[a]–2168 | 1130–2820 | 0.40–1.20[a] | 0.31–0.74 |
| 2018 | 9.3–11.0 | 7.6–10.3 | 30.7–32.6 | 30.9–33.9 | 133–209 | 65–178 | 803–1099 | 887–1338 | 0.74–0.94 | 0.62–0.86 |
| | | | | | | | | | | |
| *Admiralty Reach* | | | | | | | | | | |
| April | | | | | | | | | | |
| 2015 | 9.6–9.9 | 8.8–9.8 | 29.2–30.0 | 29.3–31.5 | 247–263 | 199–260 | 609–650 | 581–674 | 1.01–1.08 | 1.03–1.16 |
| 2016 | 9.6–9.9 | 9.3–9.8 | 28.6–29.9 | 28.7–313 | 248–279 | 221–271 | 500–544 | 528–627 | 1.15–1.19 | 1.10–1.16 |
| 2017 | 8.6–8.7 | 8.4–8.6 | 28.4–30.0 | 28.6–31.5 | 258–299 | 210–283 | 425–573 | 473–679 | 1.09–1.28 | 1.02–1.18 |
| 2018 | 8.5–8.7 | 8.2–8.7 | 29.2–30.6 | 29.3–32.6 | 243–270 | 168–265 | 709–735 | 670–884 | 0.84–0.92 | 0.85–0.93 |
| July | | | | | | | | | | |
| 2014 | 10.6–11.8 | 9.3–11.3 | 29.7–30.5 | 30.1–31.8 | 200–241 | 161–220 | 569–810 | 578–823 | 0.92–1.25 | 0.93–1.24 |
| 2015 | 11.1–13.4 | 9.3–12.7 | 29.8–30.6 | 30.1–32.1 | 192–246 | 156–228 | 512–684 | 576–790 | 1.10–1.47 | 0.98–1.32 |
| 2016 | 11.1–14.6 | 10.1–12.2 | 28.3–30.9 | 30.1–31.5 | 198–296 | 172–227 | 443–684 | 583–773 | 1.10–1.60 | 0.98–1.26 |
| 2017 | 10.8–14.3 | 9.4–12.0 | 27.4–30.5 | 29.2–31.8 | 212–349 | 153–252 | 215–559 | 437–785 | 1.28–2.48 | 0.96–1.51 |
| 2018 | 11.7–13.5 | 9.0–12.3 | 29.5–30.5 | 30.0–32.5 | 208–262 | 130–221 | 470–595 | 596–945 | 1.25–1.53 | 0.84–1.25 |
| September | | | | | | | | | | |
| 2014 | 11.1–12.5 | 9.7–12.1 | 30.3–31.2 | 30.5–32.2 | 173–192 | 139–188 | 630–879 | 756–1018 | 0.92–1.21 | 0.80–1.04 |
| 2015 | 11.3–13.1 | 10.0–12.8 | 30.6–31.5 | 30.7–32.2 | 178–210 | 149–200 | 418–848 | 745–972 | 0.96–1.76 | 0.84–1.10 |
| 2016 | 10.7–12.5 | 9.1–11.8 | 30.5–31.5 | 30.9–32.6 | 157–184 | 122–171 | 648–1363 | 791–1563 | 0.59–1.23 | 0.51–1.02 |
| 2017 | 11.1–13.6 | 9.3–12.3 | 29.7–31.1 | 30.4–32.5 | 151–202 | 118–175 | 683–930 | 799–1067 | 0.84–1.11 | 0.76–0.99 |
| 2018 | 10.8–13.0 | 9.5–12.9 | 30.3–31.4 | 30.4–32.5 | 157–196 | 128–190 | 694–870 | 765–997 | 0.92–1.15 | 0.82–1.06 |
| | | | | | | | | | | |
| *Main Basin* | | | | | | | | | | |
| April | | | | | | | | | | |
| 2015 | 9.9–10.4 | 9.4–10.0 | 28.4–31.9 | 28.8–32.8 | 235–289 | 206–260 | 554–709 | 546–827 | 0.90–1.13 | 0.79–1.14 |
| 2016 | 9.9–11.0 | 9.3–10.3 | 27.1–28.2 | 28.1–29.2 | 285–349 | 232–303 | 293–526 | 469–826 | 1.12–1.70 | 0.76–1.24 |
| 2017 | 8.4–9.2 | 8.0–8.5 | 24.0–28.3 | 28.0–29.3 | 271–385 | 242–294 | 223–658 | 534–831 | 0.87–1.70 | 0.72–1.02 |
| 2018 | 8.6–9.2 | 8.2–8.6 | 28.1–29.1 | 29.1–29.9 | 269–287 | 235–264 | 462–751 | 657–1140 | 0.82–1.23 | 0.56–0.95 |
| July | | | | | | | | | | |
| 2014 | 11.8–14.6 | 10.7–12.7 | 28.4–29.4 | 29.2–30.2 | 229–337 | 184–243 | 315–627 | 426–833 | 1.16–2.01 | 0.87–1.52 |



| | | | | | | | | | | |
|---|---|---|---|---|---|---|---|---|---|---|
| 2015 | 12.6–15.3 | 11.6–13.4 | 29.4–29.8 | 29.7–30.2 | 219–322 | 194–235 | 443–952 | 591–998 | 0.91–1.67 | 0.80–1.27 |
| 2016 | 12.9–15.5 | 11.3–12.8 | 29.1–29.6 | 29.3–30.3 | 224–347 | 183–253 | 272–566 | 495–793 | 1.29–2.43 | 0.94–1.39 |
| 2017 | 12.6–16.3 | 10.5–12.5 | 28.1–28.7 | 28.6–29.6 | 261–419 | 173–265 | 155–502 | 416–842 | 1.34–3.45 | 0.82–1.54 |
| 2018 | 13.0–15.7 | 11.1–13.2 | 28.5–29.5 | 29.5–30.2 | 243–397 | 166–242 | 404–865 | 578–1133 | 0.90–1.79 | 0.67–1.28 |
| September | | | | | | | | | | |
| 2014 | 12.8–13.9 | 11.8–13.1 | 29.2–30.3 | 30.1–30.9 | 188–362 | 156–220 | 271–697 | 567–960 | 1.14–2.33 | 0.84–1.35 |
| 2015 | 13.6–14.3 | 12.3–13.8 | 30.4–30.5 | 30.4–31.1 | 197–304 | 165–227 | 379–780 | 495–1084 | 1.09–1.96 | 0.79–1.57 |
| 2016 | 13.2–14.0 | 12.0–13.3 | 30.2–30.3 | 30.3–30.8 | 177–204 | 148–190 | 711–844 | 821–990 | 0.97–1.18 | 0.82–1.00 |
| 2017 | 13.2–14.6 | 12.0–13.3 | 29.7–29.9 | 29.9–30.5 | 183–241 | 152–191 | 687–1540 | 770–1887 | 0.56–1.14 | 0.45–1.02 |
| 2018 | 13.1–14.4 | 12.2–13.3 | 29.9–30.3 | 30.3–30.7 | 182–239 | 157–191 | 577–853 | 765–968 | 0.97–1.39 | 0.84–1.06 |
| | | | | | | | | | | |
| *South Sound* | | | | | | | | | | |
| April | | | | | | | | | | |
| 2015 | 10.2–11.8 | 9.8–10.3 | 28.0–31.6 | 28.3–31.8 | 255–417 | 194–265 | 169–609 | 617–964 | 1.02–2.92 | 0.70–1.05 |
| 2016 | 9.8–13.5 | 9.5–10.0 | 27.3–27.8 | 27.6–28.2 | 278–432 | 241–294 | 171–524 | 535–774 | 1.09–2.82 | 0.77–1.06 |
| 2017 | 8.5–9.2 | 8.3–8.5 | 27.1–27.9 | 27.7–28.3 | 264–315 | 256–267 | 445–706 | 651–769 | 0.78–1.20 | 0.74–0.87 |
| 2018 | 8.9–9.4 | 8.4–8.9 | 28.0–28.5 | 28.6–29.0 | 283–386 | 248–304 | 247–659 | 496–814 | 0.90–2.02 | 0.75–1.16 |
| July | | | | | | | | | | |
| 2014 | 13.1–15.6 | 12.2–13.9 | 28.6–29.0 | 28.8–29.3 | 249–304 | 217–260 | 351–556 | 468–685 | 1.29–2.04 | 1.06–1.53 |
| 2015 | 14.1–15.2 | 12.9–14.6 | 29.1–29.4 | 29.3–29.6 | 241–278 | 202–252 | 480–590 | 554–786 | 1.32–1.58 | 0.99–1.41 |
| 2016 | 13.5–15.2 | 12.7–13.8 | 29.0–29.3 | 29.2–29.6 | 228–282 | 191–229 | 332–589 | 612–746 | 1.27–1.98 | 1.00–1.22 |
| 2017 | 13.0–15.9 | 11.7–13.1 | 28.2–28.5 | 28.5–28.8 | 244–347 | 188–246 | 322–566 | 180–788 | 1.21–2.10 | 0.88–2.69 |
| 2018 | 13.2–16.5 | 12.6–13.7 | 29.0–29.2 | 29.3–29.6 | 236–411 | 196–247 | 267–595 | 587–822 | 1.23–2.52 | 0.91–1.26 |
| September | | | | | | | | | | |
| 2014 | 13.6–15.2 | 13.0–13.9 | 29.3–29.9 | 29.6–30.1 | 185–333 | 167–208 | 356–805 | 501–792 | 1.02–2.02 | 1.03–1.52 |
| 2015 | 14.3–15.1 | 13.8–14.9 | 30.0–30.2 | 30.0–30.4 | 136–206 | 159–202 | 690–1069 | 715–1041 | 0.83–1.24 | 0.84–1.20 |
| 2016 | 13.9–14.7 | 13.3–14.1 | 29.3–30.0 | 30.0–30.3 | 176–235 | 154–192 | 680–985 | 802–1033 | 0.85–1.20 | 0.81–1.03 |
| 2017 | 13.6–15.3 | 13.2–13.4 | 29.1–29.4 | 29.4–29.8 | 160–255 | 169–202 | 830–2070 | 984–1564 | 0.45–1.04 | 0.55–0.89 |
| 2018 | 13.7–14.7 | 13.4–14.2 | 29.6–29.9 | 29.8–30.1 | 168–288 | 173–209 | 471–863 | 647–850 | 0.96–1.63 | 0.97–1.24 |
| | | | | | | | | | | |
| *Whidbey Basin* | | | | | | | | | | |
| April | | | | | | | | | | |
| 2015 | 10.2–11.3 | 9.5–10.1 | 20.4–28.5 | 28.6–29.5 | 217–410 | 172–268 | 146–778 | 704–1156 | 0.77–1.97 | 0.59–0.92 |
| 2016 | 9.6–11.6 | 9.3–10.1 | 21.5–27.6 | 27.7–29.2 | 242–376 | 180–285 | 140–714 | 463–1173 | 0.81–2.09 | 0.56–1.16 |
| 2017 | 8.6–9.4 | 8.2–8.6 | 20.7–26.8 | 26.2–29.3 | 306–389 | 223–296 | 110–368 | 463–922 | 1.26–2.13 | 0.66–1.14 |
| 2018 | 8.6–9.6 | 8.3–9.1 | 24.5–28.8 | 28.8–29.8 | 204–294 | 175–263 | 549–1072 | 781–1218 | 0.59–0.86 | 0.54–0.79 |
| July | | | | | | | | | | |
| 2014 | 10.1–17.7 | 9.1–12.2 | 22.0–28.9 | 28.8–30.0 | 217–454 | 159–248 | 113–763 | 500–1405 | 0.86–2.94 | 0.49–1.36 |
| 2015 | 11.9–18.6 | 10.4–12.7 | 25.9–29.6 | 29.3–30.2 | 224–415 | 134–231 | 183–667 | 583–1366 | 1.06–2.67 | 0.53–1.28 |
| 2016 | 10.6–17.6 | 10.4–11.7 | 22.1–29.4 | 28.8–30.4 | 168–458 | 146–195 | 269–1212 | 759–1278 | 0.42–2.02 | 0.55–0.96 |
| 2017 | 9.9–15.0 | 9.2–11.8 | 22.9–28.4 | 28.5–29.6 | 172–384 | 145–227 | 143–947 | 569–1353 | 0.67–2.55 | 0.48–1.17 |
| 2018 | 11.2–17.9 | 9.7–12.3 | 22.3–29.5 | 29.2–30.1 | 204–441 | 140–206 | 119–847 | 777–1377 | 0.81–3.12 | 0.51–0.95 |
| September | | | | | | | | | | |
| 2014 | 12.1–14.2 | 11.0–12.4 | 25.6–30.0 | 29.6–30.8 | 157–258 | 93–176 | 480–1007 | 685–1581 | 0.72–1.27 | 0.50–1.14 |
| 2015 | 12.7–15.2 | 12.2–13.7 | 24.3–30.4 | 30.0–31.1 | 101–365 | 89–252 | 210–1529 | 825–1609 | 0.54–2.15 | 0.52–1.03 |
| 2016 | 12.3–14.9 | 11.7–12.8 | 25.4–30.3 | 29.7–30.9 | 137–380 | 97–170 | 180–1135 | 890–1549 | 0.67–2.69 | 0.51–0.91 |
| 2017 | 11.3–15.1 | 10.9–12.9 | 27.4–29.6 | 29.2–30.5 | 146–259 | 107–179 | 514–1324 | 786–1437 | 0.55–1.37 | 0.51–0.99 |
| 2018 | 12.4–14.6 | 11.0–13.4 | 26.8–30.1 | 29.5–30.7 | 200–379 | 100–204 | 188–1008 | 730–1692 | 0.76–2.78 | 0.45–1.11 |
| | | | | | | | | | | |
| *Hood Canal* | | | | | | | | | | |
| April | | | | | | | | | | |
| 2015 | 10.2–12.2 | 9.7–10.6 | 24.1–29.0 | 29.0–30.2 | 201–481 | 46–321 | 113–879 | 369–2439 | 0.75–3.15 | 0.32–1.60 |
| 2016 | 10.0–12.4 | 9.6–11.8 | 21.8–28.6 | 28.4–30.5 | 268–413 | 64–293 | 148–1800 | 416–2100 | 0.41–2.60 | 0.38–1.37 |
| 2017 | 8.6–11.3 | 8.5–10.5 | 15.6–29.2 | 28.1–30.3 | 225–467 | 94–284 | 68–886 | 375–1747 | 0.70–2.42 | 0.43–1.42 |
| 2018 | 9.0–11.1 | 8.4–10.5 | 25.5–29.7 | 29.1–30.6 | 142–282 | 98–269 | 538–1597 | 614–1812 | 0.47–1.12 | 0.44–1.03 |
| July | | | | | | | | | | |
| 2014 | 10.9–20.9 | 8.4–12.3 | 25.8–29.8 | 29.1–30.2 | 230–386 | 84–258 | 231–709 | 456–2274 | 1.02–2.74 | 0.31–1.54 |
| 2015 | 12.0–21.8 | 10.2–13.0 | 26.1–30.1 | 29.6–30.6 | 199–388 | 12–247 | 220–1450 | 478–3460 | 0.57–2.87 | 0.23–1.57 |
| 2016 | 11.2–17.9 | 10.7–12.3 | 26.9–30.0 | 29.4–30.7 | 89–327 | 22–210 | 252–2185 | 615–2814 | 0.39–2.57 | 0.28–1.21 |
| 2017 | 11.5–21.1 | 9.1–12.4 | 20.9–29.0 | 28.7–30.0 | 223–394 | 63–255 | 223–940 | 408–2259 | 0.77–2.56 | 0.34–1.62 |
| 2018 | 11.2–21.8 | 9.5–13.8 | 25.0–29.9 | 29.4–30.8 | 99–474 | 63–246 | 212–1625 | 466–2309 | 0.48–2.85 | 0.33–1.57 |
| September | | | | | | | | | | |
| 2014 | 9.3–15.4 | 8.8–11.9 | 24.1–30.5 | 29.6–31.2 | 66–389 | 62–180 | 292–2611 | 665–2475 | 0.28–2.02 | 0.30–1.16 |
| 2015 | 12.0–15.3 | 11.8–13.0 | 27.0–30.5 | 30.3–31.3 | 79–259 | 88–213 | 707–1822 | 686–1755 | 0.46–1.14 | 0.48–1.15 |
| 2016 | 11.4–15.0 | 11.4–12.2 | 26.3–30.4 | 30.1–31.2 | 53–259 | 75–176 | 293–2120 | 919–1769 | 0.38–2.01 | 0.46–0.88 |
| 2017 | 9.7–17.5 | 9.6–12.1 | 24.5–30.2 | 29.4–31.0 | 106–317 | 58–175 | 285–3122 | 460–3372 | 0.25–2.29 | 0.23–1.47 |
| 2018 | 10.1–14.5 | 9.8–12.3 | 28.0–30.2 | 29.8–31.0 | 46–291 | 49–181 | 156–2641 | 789–2612 | 0.29–2.93 | 0.29–1.02 |




### 4.2 Biogeochemical seasonality across basins

### 4.2.1 Dissolved oxygen

Wide ranges of oxygen content were seen at coastal and SJdF stations, at surface and subsurface depths. Deep-water oxygen
observations frequently had higher variability than surface waters, with strong interannual variability in $O_2$ medians, which
were all above the hypoxia threshold (i.e., 62 µmol kg$^{-1}$ = 2.0 mg L$^{-1}$ = 1.5 mL L$^{-1}$), and ranges that occasionally dipped into
hypoxic conditions (Figure 6A, Table 4). No clear seasonal difference in deep $O_2$ content emerged between early and late in
the upwelling season, but deep-water $O_2$ medians were lower during O17, M18, and O18 in boundary waters. Boundary water
surface $O_2$ medians were consistently higher than subsurface values during any single cruise, although ranges sometimes
overlapped across depths. Surface ranges were wider at SJdF stations, but subsurface ranges were widest at the coastal stations.

Oxygen content in deep AR waters decreased in median, minimum, and maximum values from April to September each year
(Figure 6B). Surface $O_2$ also showed decreasing median values April–September, but surface ranges were narrower than at
depth in spring and fall, such that surface $O_2$ observations ranges in April and September did not overlap. Wider $O_2$ ranges and
higher outliers were observed during J16 and J17 at the surface, and wider ranges with low outliers were seen at depth during
A17, J17, A18, and J18.

Open waters of MB and SS were consistently well-oxygenated to the bottom (Figure 6B, Table 4). MB bottom water oxygen
was >220 µmol kg$^{-1}$ in spring and declined to median values of ~160–180 µmol kg$^{-1}$ during September surveys (Figure 6B,
*cf*. Figure 9 in Alin et al., 2023a).  Deep waters in SS only fell below 180 µmol kg$^{-1}$ twice, with minimum values of <160 µmol
kg$^{-1}$ during S16 and S17. $O_2$ content in MB and SS surface waters was always >200 µmol kg$^{-1}$ during April and July cruises,
and occasionally medians were >300 µmol kg$^{-1}$ (e.g., A16 in both basins and A17 and J17 in MB). Across seasons, variability
was higher in surface than deep waters, with generally decreasing median values throughout the water column from April to
September.


Surface waters in WB and HC had wide ranges of $O_2$ through spring–fall cruises, typically with lower median $O_2$ content in
Septembers compared with Aprils or Julys, particularly in HC (Figure 6B). Subsurface $O_2$ variability was lower in WB,  but
$O_2$ variability remained high in HC deep water (Table 2). A progressive April–September decline in subsurface $O_2$ medians
was observed in WB, although in HC variability declined more consistently than medians across seasons. Lower surface $O_2$
medians were observed during J16, A18, and J18 in WB and HC, with higher medians in A15, S15, A16, and S18 in WB.
Deep WB waters had higher median values in A17 and high outliers in S15. In deep HC waters, median $O_2$ values in A17
appear higher than normal, while J15 and J16 $O_2$ medians and minima were ~25–50 µmol kg$^{-1}$ lower than other Julys. The





only measurements of hypoxic conditions in Puget Sound were taken in HC, at depth during S14, A15, J15, J16, S17, and S18 cruises, and in surface waters during S16 and S18.




**Figure 6: A)** Raincloud plots for adjusted CTD oxygen in coastal and Strait of Juan de Fuca surveys in the early and late upwelling season beginning in the fall of 2014. Figure organization is the same as in Fig. 3A. **B)** Raincloud plots for adjusted CTD oxygen in Puget Sound surveys in April, July, and September beginning in July 2014. Figure organization is the same as in Fig. 3B.






Looking across basins, surface oxygen content occupied similar overall range widths in spring and fall, while medians declined seasonally by 35–125 µmol kg$^{-1}$ everywhere (Figure 5, Table 2). Surface $O_2$ variability was highest across seasons in HC, WB, and SJdF and lowest in AR. Surface variability was highest in summer in PS basins except SS where it was lowest. WB surface $O_2$ medians and range width peaked and were highest among PS basins in summer, approaching spring coastal surface $O_2$

median values. Everywhere else, surface $O_2$ medians decreased from spring to fall. At depth, $O_2$ content decreased monotonically by 52–94 µmol kg$^{-1}$ across all PS basins from April to September, with the largest decrease at AR and the smallest decrease across seasons in HC. In contrast, median $O_2$ in deep boundary waters remained roughly the same from spring to fall, with variability often exceeding surface variation. Ranges in deep $O_2$ content were widest in HC during spring–summer, followed by coastal and SJdF stations. While we observed hypoxic conditions in surface waters and near-anoxia at

depth in HC, $O_2$ concentrations below the hypoxia threshold have not been observed elsewhere in PS during these cruises. $O_2$ concentrations were consistently second lowest at the river end of the WB basin, with lowest $O_2$ conditions occurring consistently in September, even during heatwave years when deep HC $O_2$ was lowest during Julys.

### 4.2.2 Carbon dioxide fugacity (*f*CO₂)

Subsurface $CO_2$ fugacity values typically had lower highs, lows, and medians at coastal than SJdF stations (Figures 7A and

S4, Table 2). Coastal surface median values were thus often undersaturated with respect to atmospheric *f*CO₂, whereas SJdF medians were often above atmospheric values. As for $O_2$, no clear seasonal *f*CO₂ difference was evident, either at the surface or at depth. Coastal stations had higher surface median *f*CO₂ in M15, but the most notable variation was the anomalously wide *f*CO₂ ranges observed during O17, with deep-water medians >1000 µatm at coastal stations (high=2376 µatm) and ~1900 µatm at SJdF stations (high=2820 µatm). O17 SJdF surface *f*CO₂ values were unprecedented, with a median of 1575 µatm and highs

up to 2168 µatm. Coastal surface *f*CO₂ anomalies were also notable during O17, with the only observations of *f*CO₂ >1000 µatm occurring then. SJdF M18 subsurface and O18 median *f*CO₂ observations across depth were also somewhat elevated compared to other cruises, suggesting possible carryover of the anomalously acidified water masses from the previous fall.

AR *f*CO₂ medians and ranges increased between April and September, with most cruises having narrow ranges and little depth

structure (Figure 7B, S5, Table 2). The relatively wide overall AR *f*CO₂ ranges reflect low surface outliers in J17 and high outliers across depths in S16. The latter co-occurred with the lowest $O_2$ median in AR surface waters in this cruise time-series (*cf.* Figure 6B).

In MB and SS, *f*CO₂ variability across depths was lower during April–July than September (Figures 7B, S5, Table 2). The

majority of MB-SS observations >1000 µatm were associated with extremely high *f*CO₂ anomalies in S17. Extreme *f*CO₂ conditions in S17 were preceded by anomalously low *f*CO₂ in MB surface waters in J17 and to a lesser extent in A17, which



we interpret as reflecting a protracted season of biological drawdown due to the co-occurrence of high $O_2$, low $f$CO$_2$, and sustained high chlorophyll (PSEMP Marine Waters Workgroup, 2018).



**Figure 7: A)** Raincloud plots for the fugacity of carbon dioxide in coastal and Strait of Juan de Fuca surveys in the early and late upwelling season beginning in the fall of 2014. Figure organization is the same as in Fig. 3A. **B)** Raincloud plots for the fugacity of carbon dioxide in Puget Sound surveys in April, July, and September beginning in July 2014. Figure organization is the same as in Fig. 3B.

Variability in $f\text{CO}_2$ was highest across regions in HC and second highest in WB, with similar range widths in surface and subsurface waters in both basins (Figures 7B, S5, Table 2). Surface $f\text{CO}_2$ medians tended to be lower in Aprils and Julys in both basins. HC had higher high $f\text{CO}_2$ values across depths and months than WB. Cruise medians were within ±230 µatm between WB and HC surface waters in Aprils and Julys, but HC surface water medians were more supersaturated with $f\text{CO}_2$ in all Septembers except S14. HC surface waters had particularly high $f\text{CO}_2$ outliers during S14, S17, and S18, with the highest deep water HC values seen during J15 and S17. WB surface $f\text{CO}_2$ had the highest outliers during S15 and an anomalously high median during A18, which could reflect persistence of acidified conditions from fall 2017.

The majority of the water column was supersaturated with respect to atmospheric $f\text{CO}_2$ values (i.e., >400 µatm) in most places and times for the duration of this time-series (Tables 4–5), reflecting the importance of respiration processes in Salish Sea $\text{CO}_2$ chemistry, although moored time-series show that surface undersaturation prevails in spring–summer (Alin et al. in PSEMP Marine Waters Workgroup, 2021; Fassbender et al., 2018). Average surface $f\text{CO}_2$ medians and ranges were lower and narrower, respectively, during spring and summer months than fall across regions. As for temperature, salinity, and oxygen, HC, WB, and SJdF had the largest surface variability (Figure 5). Median surface $f\text{CO}_2$ values were lowest across coastal, WB, and HC stations in spring, and in HC and WB among PS basins in summer. The highest spring surface median values were in SJdF and AR, reflecting vigorous mixing bringing deep $\text{CO}_2$-rich waters to the surface. In fall, HC had comparably high surface median $f\text{CO}_2$ values to SJdF and AR, with the highest variability in HC and SJdF. In deep water, median $f\text{CO}_2$ values were higher than at the surface across seasons by ~30–400 µatm at AR, MB–SS, and boundary water stations, and by ~350–725 µatm in WB and HC. Surface and deep $f\text{CO}_2$ levels overlapped most at AR in all seasons and in HC during fall, when local upwelling brings deep water enriched in $f\text{CO}_2$ to near-surface depths as it is flushed out of the basin via estuarine circulation (Figures 7 and S5). Median deep $f\text{CO}_2$ values were highest in HC across seasons, although SJdF and WB approached HC levels at depth during fall cruises. Coastal deep median $f\text{CO}_2$ values were comparable to MB and SS and higher than AR values in spring but lowest across regions in fall.

### 4.2.3 Aragonite saturation state ($\Omega_{arag}$)

Both surface and subsurface boundary water $\Omega_{arag}$ spanned undersaturated to quite supersaturated (> 1) values, with strong surface variability and lower highs and lows in SJdF (Figure 8A, Table 2). Spring surface $\Omega_{arag}$ ranges were wider in both regions than fall ranges, with more variable median values, particularly at the coast. The highest deep medians occurred in O14 in SJdF and O16 at coastal stations, and the lowest medians were seen in O17 in both regions. Notably high surface median $\Omega_{arag}$ values were observed in M16 and M18 at coastal stations and M17 at SJdF stations, with notable surface low medians during O17 at both coastal and SJdF stations, as well as during M15 at coastal stations.









**Figure 8: A)** Raincloud plots for aragonite saturation state in coastal and Strait of Juan de Fuca surveys in the early and late upwelling season beginning in the fall of 2014. Figure organization is the same as in Fig. 3A. **B)** Raincloud plots for aragonite saturation state in Puget Sound surveys in April, July, and September beginning in July 2014. Figure organization is the same as in Fig. 3B.

In contrast to $O_2$ and $f$CO$_2$, surface $\Omega_{arag}$ medians at AR tended to be highest in Julys, followed by Aprils, with Septembers having the lowest values. The same pattern was evident but weaker at depth at AR. Ranges for $\Omega_{arag}$ were typically widest in July (Table 4). Anomalously wide ranges in $\Omega_{arag}$ were seen in surface waters with high anomalies in S15 and J17 and with a lower median and lows in S16.

Whereas $f$CO$_2$ variability was similar across depths in MB and SS, $\Omega_{arag}$ generally had substantially higher variability in surface waters than deep (Figure 8B, Table 2). Both surface and subsurface MB and SS waters had highest $\Omega_{arag}$ medians in Julys, with lower medians and significant interannual variability across Aprils and Septembers. Median deep SS $\Omega_{arag}$ observations were above saturation during Julys and some Septembers, whereas MB subsurface medians were typically below the saturation threshold ($\Omega_{arag} = 1$), reflecting the greater volume of deep water in MB. Exceptions occurred during J14, J15, and J16, when MB deep waters had medians >1, and J17 in SS, which had high deep-water outliers. Median surface $\Omega_{arag}$ values were supersaturated for all cruises except A18 in MB, A17 in SS, and S17 in both basins. Notably high $\Omega_{arag}$ outliers in surface waters were observed during A15, A16, and J18 in SS and J17 in MB, which also had the highest median observed across all basins and seasons in PS. Notably low $\Omega_{arag}$ medians occurred in S17 in both basins and across depths.

In WB and HC, $\Omega_{arag}$ ranges were much wider in surface than subsurface waters, with wider ranges at both depths in HC than WB (Table 4). $\Omega_{arag}$ ranges tended to be widest in WB surface waters in July, with no clear seasonal pattern in surface medians (Figure 8B). Notable WB surface $\Omega_{arag}$ medians were >2 in J14, J15, and S18 and <1 in A18, J16, and S17. Surface HC $\Omega_{arag}$ medians were notably low in A18 and J16 and high in S14 (Figures 8B and S6). Deep-water $\Omega_{arag}$ medians were more stable across months and consistently undersaturated.

In summary, median aragonite saturation states were substantially higher in surface coastal waters in spring and fall than throughout SJdF and PS basins (Figure 5). HC had the highest surface variability across seasons, with the highest PS medians in spring and summer and the lowest in fall. AR had the least inter- and intra-seasonal variability of all regions in both surface and deep water. Deep HC medians were lowest across months, with consistently high variability reflecting considerable along-basin gradients at depth (Figure S6). Deep WB $\Omega_{arag}$ medians fell between those for PS and HC, but the range widths were more similar to PS basins than HC. Notably low $\Omega_{arag}$ anomalies occurred in fall 2017, with indications that acidified conditions were held over until A18 observed in multiple basins.



### 4.3 The role of distinct seasonality across parameters and basins in driving severity of acidification and hypoxia

Average ocean conditions from the coast through Puget Sound are summarized in bubble plots of each parameter for each month, region, and depth across all 2014–2018 cruises (Figure 5). Seasonal variation of median values and ranges across basins was not consistent across parameters. For instance, surface salinity seasonality was different from temperature seasonality across PS basins. Surface water temperatures peaked in some PS basins in summer, following solar radiation (Figure 2), while deep-water temperatures continued to rise until fall, except at AR (Figure 3B). In contrast, PS salinities progressively increased and ranges contracted from spring to fall, tracking seasonal precipitation and river discharge patterns (Figure 4B, references in Table 1, and Banas et al., 2015). The continued increase of deep temperature and salinity until fall reflects a combination of surface conditions mixing to depth and the influence of upwelling bringing colder, saltier, lower oxygen water masses into the Salish Sea, and displacing the fresher, warmer, more oxygenated water masses that are present in MB and SS in winter. Moored time-series provide a longer-term, more temporally resolved context for the seasonal variability across parameters and PS basins observed in the Salish cruise time-series and confirm that water mass properties do not vary consistently across PS basins. Specifically, HC deep-water seasonal peaks and troughs for temperature lag those from other basins by a few months (https://nwem.apl.washington.edu/prod_PS_ClimateTrends.shtml). Salinities reach their peaks and nadirs 1–2 months after temperature in both surface and deep waters across all PS moorings.

$O_2$ medians in most basins and across depths declined steadily from spring to fall, whereas $\Omega_{arag}$ medians across depths usually peaked in summer and $f$CO$_2$ levels typically increased most substantially by fall (Figure 5). Variability in $O_2$ and $\Omega_{arag}$ was markedly lower in subsurface than surface waters, although deep-water HC ranges were still wide (Figures 6, 8, Table 2). However, subsurface $f$CO$_2$ ranges were typically as wide as surface ranges, and in the case of HC, often wider (Figure 7, Table 2), which likely reflects the amplification of $f$CO$_2$ variability that occurs when buffering capacity declines (Pacella et al., 2018; Kwiatkowski and Orr, 2018).

Oxygen and inorganic CO$_2$ dynamics often mirror each other within a local water mass because they are linked by the stoichiometry of biological production and respiration processes, but these can be decoupled across depth (e.g., Cai et al., 2011; Feely et al., 2010). Surface $f$CO$_2$ and $O_2$ levels dominantly reflect phytoplankton bloom activity, which peaks from spring to summer throughout the study region (e.g., PSEMP Marine Waters Workgroup, 2019 and earlier years). Organic matter from phytoplankton blooms subsequently drives the regional spring-to-fall $O_2$ decrease and $f$CO$_2$ increase via respiration in both surface and subsurface waters. While CO$_2$ drawdown also affects saturation states, $\Omega_{arag}$ peaked in summer in many basins, reflecting stronger temperature influence on carbonate ion concentrations than $f$CO$_2$ in surface waters (Figure 1 in Cai et al., 2020). In deep water masses, biogeochemical parameters tended to follow more monotonic seasonal trajectories, with T, S, and $f$CO$_2$ increasing and $O_2$ decreasing across spring–fall as a result of longer residence times and respiration contributing to higher $f$CO$_2$ (e.g., Feely et al., 2010, 2023). Thus, seasonal decoupling across metrics of ocean acidification and oxygenation



reflects the relative importance of physical versus biological control on each parameter, which have strong gradients across
565 the estuarine to coastal to open ocean continuum (Kwiatkowski and Orr, 2018; Lowe et al., 2019; Cai et al., 2020).

Interpreting Salish cruise seasonal patterns in the context of the moored climatologies across PS shows that deep climatological temperature and salinity minima tend to co-occur with maximum annual deep $O_2$ values and vice versa from AR to SS, whereas annual low and high values of T, S, and $O_2$ occur synchronously in HC. The lag in temperature and salinity seasonality in deep
570 HC waters is consistent with the longer residence and transit times along the deep axis of HC compared to other basins (Babson et al., 2006; MacCready et al., 2021). While deep-water renewal in MB, SS, and WB basins follows mixing of incoming upwelled water with outgoing surface water at AR, deep-water renewal in HC requires incoming marine waters to pass over a second glacial sill before transport along the long deep axis of HC, contributing to lags in temperature and salinity seasonality in deep HC waters. Consequently, mid-summer to fall bottom water in HC was often colder and fresher than in MB–SS, while
575 in winter–spring, bottom water toward the southern end of the HC basin was frequently warmer and saltier (Figures 5–6 in Alin et al., 2023a). Peak surface $O_2$ values occur earliest in HC, followed by SS and AR, with MB peaking latest, spanning spring to summer. Climatological low surface $O_2$ values span fall to winter, occurring earliest at AR, followed by MB, SS, and finally HC. Collectively, these observations suggest that the earlier surface $O_2$ peak ($f$CO$_2$ nadir) in HC surface waters, which along with higher chlorophyll values imply earlier seasonal onset and peak of primary production (PSEMP Marine Waters
580 Workgroup, 2018), translates to earlier $O_2$ depletion in deep waters driven by remineralization of sinking phytoplankton. The earlier peaks in production and respiration in HC thus effectively offset the deep-water lag in $O_2$ minimum (and $f$CO$_2$ maximum) timing relative to T and S seasonality seen in other basins. This is important because the interaction and timing of seasonal peaks in physical and biogeochemical processes drives in the formation of hypoxic, corrosive conditions in HC deep waters. Thus, future changes in deep-water renewal timing—as observed during the MHW—and the phenology of surface
585 biological processes may influence deep-water conditions differently across the distinct basins of this complex ecosystem and could result in new acidification or hypoxia hotspots emerging.

## 5 Discussion: How did major environmental anomalies of 2013–2018 affect physical and biogeochemical conditions?

The 2014–2016 part of the Salish cruise time-series coincided with the protracted, intense heat anomaly initiated by the 2013–
2015 NE Pacific marine heatwave (MHW) and extended by the 2015–2016 El Niño event (Bond et al., 2015; Jacox et al.,
2016; Gentemann et al., 2017; Peterson et al., 2017). The combination of the Pacific Ocean heat anomalies and a protracted atmospheric heat anomaly affecting the western U.S. during 2014–2015 (Swain et al., 2016) altered atmospheric and seawater temperatures in the Salish Sea (Table 1 and references therein). Record precipitation occurred during at least one month per year throughout 2014–2018, with river discharge timings showing strong anomalies tending toward earlier, higher peak runoff and lower summer flows than in the historical baseline (Table 1 and references therein). Below we connect how the timing of
these environmental anomalies contributed to the oceanographic anomalies observed in Washington's marine waters, focusing





on understanding the genesis of summer and fall deep-water anomalies, because the strongest hypoxic and acidified conditions typically occur then. We focus on the oceanographic responses of boundary waters, weakly stratified MB, and strongly stratified HC to examine whether the 2013–2018 environmental anomalies caused changes in timing or decoupling of physical or biogeochemical processes such as those described in Section 4.3, and in so doing, worsened or ameliorated ocean
acidification and hypoxia at new locations or times.

### 5.1 Physical oceanographic anomalies during 2014–2018

Temperature and salinity anomalies observed in the 2014–2018 Salish cruise time-series did not always occur synchronously in space or time. The intense NE Pacific marine heat anomalies manifested as surface temperature medians in O14 that were mostly outside the envelope of other 2014–2018 boundary water temperatures (Figure 3). At depth, the widest boundary water
temperature ranges were observed in O14. Wider deep-water ranges and higher medians persisted through 2015–2016. A larger-than-normal volume of warm coastal surface waters was also observed during the O16 cruise (Figure 4 in Alin et al., 2023a), but temperatures were lower than O14. Nearby coastal time-series showed that warm anomaly events of up to 4.5 °C lasting for 10–20 days and affecting the water column to bottom depths of 40 m during 2014 and 2015 (Koehlinger et al., 2023).


Comparison of summer–fall PS cross-sections before and after the arrival of heatwave-warmed waters on the coast in September 2014 (Peterson et al., 2017; Koehlinger et al., 2023) provides perspective on the timeline of MHW warming across PS basins (Figure 5 in Alin et al., 2023a). Within PS basins, surface warming reflected regionally elevated air temperatures and greater than average solar radiation (Table 1), with surface warm extremes of ≥21° C during J15, J17, and J18 in southern
HC being >2.5° C above monthly averages from Fassbender et al. (2018) (Figure 5 in Alin et al., 2023a, Tables 2–3). Peak MHW surface temperatures were cooler overall in MB, but still 1.8–3.1°C above average (Fassbender et al., 2018). Subsurface water temperatures did not reflect the strong MHW warming seen on the coast during 2014, but temperature increases were observed throughout the water column in most basins throughout 2014–2016, with some deep-water warming evident through 2018 (*cf.* August–October cruises 2008–2014 in Figure 5). For instance, deep waters in southern HC warmed by ~2 °C during
Julys and Septembers of 2015–2016 relative to the same months in 2014, subsequently cooling by ~1 °C in 2017–2018. A deep mixing event in HC during anomalously cold weather in February 2014 caused cooler deep water to persist in HC through 2014; thus, it is unclear whether deep HC September temperatures in 2017–2018 reflect a return to normal (PSEMP Marine Waters Workgroup, 2015). For comparison, heat from the NE Pacific marine heatwave of 2013–2015 lingered at depth until at least early 2018 in a fjord to the north of the Salish Sea, indicating pronounced persistence of the MHW warming signature
in other deep, stratified basins in the region (Jackson et al., 2018).

Boundary water observations in O14 reflected a fresher water column at the time when the MHW arrived in these waters than during any other late upwelling season cruise in this time-series (Table 2), consistent with a warmer, fresher water mass from





the NE Pacific being advected to the Washington and Oregon coastline in fall 2014 (Peterson et al., 2017; Koehlinger et al.,
2023). Octobers salinities during 2014–2016 remained fresher by ~1, overlapping more with spring salinities than O17 and
O18 and illustrating the longevity of the fresher water masses associated with the MHW at the coast. The combination of
higher temperatures and lower salinities in boundary waters during O14 and O16 manifested as substantially less dense waters
to 100 m depth (Figure S4).

Within PS, salinity anomalies related to precipitation and winter–spring river discharge anomalies should manifest as
freshening anomalies during spring. The lowest surface salinities were observed in WB and HC during Aprils of 2015–2017,
and Julys of 2014 and 2017–2018 (Figure 6 in Alin et al., 2023a, Table 2). Lower A17 and J17 salinities in MB and HC reflect
the wettest February–April on record and strong PS river runoff January–March (Figure 4B, Tables 1–2), with salinities two
standard deviations below average also observed in PS moored time-series (Ruef et al. in PSEMP Marine Waters Workgroup,
2018). Lower salinities persisted throughout the water column in all PS basins across seasons in 2017 (Figure 6 in Alin et al.,
2023a). Spring 2017 surface salinities were fresher in boundary waters and at depth on the Washington shelf (the latter
observed by Mickett and Newton in PSEMP Marine Waters Workgroup, 2018), but the freshening signature had disappeared
from boundary waters by O17. Other data sources show that the Salish Sea experienced one of the top five annual total
freshwater inflow years since 1999 during 2017 (Khangaonkar et al., 2021).


Conversely, summer drought conditions increased in severity and duration across 2015, 2017, and 2018 (Table 1 and references
therein) and could be reflected by higher-than-normal salinity anomalies during September cruises (Figure 6 in Alin et al.,
2023a). The lowest surface fall salinities in WB and HC were observed in S14 and S16, whereas in drought years, minimum
surface salinities were ~1–2 salinity units higher in September (Table 2).  Compared to monthly average surface salinities from
moorings, the lowest salinity values seen in HC in S15 and S17 were up to 2.5 salinity units higher than average (Fassbender
et al., 2018). In MB, the persistent fresher conditions (by ~0.5 salinity unit) in the upper water column during S17 suggest that
low river discharges during the summer 2017 drought may have slowed estuarine circulation, allowing the fresher, more
stratified conditions from earlier in 2017 to persist (Table 1; *cf.* Fassbender et al., 2018).




**Table 3.** Distribution of temperature (T), oxygen (O₂), calcite and aragonite saturation states ($\Omega_{calc}$, $\Omega_{arag}$), and carbon dioxide fugacity ($f$CO₂) observations in the Salish cruise data package relative to thresholds with potential implications for altering carbon cycle fluxes or affecting physiological processes or survival in Salish Sea species. Boundary waters includes both coastal and Strait of Juan de Fuca stations. The total number of observations (Total $n$), number of surface observations (Surface $n$, ≤20 dbar), and percent of each parameter's observations (%) in the Salish cruise data package crossing the threshold for each parameter are given in the three respective righthand columns, with total observations by basin broken down in columns to the left.

| Threshold | Boundary waters | Admiralty Reach | Main Basin | South Sound | Whidbey Basin | Hood Canal | Total $n$[a] | Surface $n$ | % |
|---|---|---|---|---|---|---|---|---|---|
| T ≥ 21 °C[b] | — | — | — | — | — | 9 | 9 | 9 | 0.1 |
| T ≥ 19 °C[b] | — | — | — | — | — | 38 | 38 | 38 | 0.5 |
| T ≥ 15 °C[g] | 28 | 1 | 10 | 25 | 31 | 114 | 209 | 203 | 2.8 |
| O₂ ≤ 62 μmol kg⁻¹[(c)] | 36 | — | — | — | — | 64 | 100 | 19 | 1.3 |
| O₂ ≤ 110 μmol kg⁻¹[(b,c)] | 220 | 2 | — | — | 26 | 386 | 634 | 76 | 8.4 |
| O₂ ≤ 155 μmol kg⁻¹[(b,c)] | 471 | 38 | 15 | 5 | 165 | 925 | 1624 | 181 | 21.7 |
| $\Omega_{calc}$ < 1 [d] | 70 | 9 | 13 | 17 | 78 | 441 | 628 | 131 | 14.3 |
| $\Omega_{arag}$ < 1 [d] | 599 | 330 | 444 | 226 | 234 | 873 | 2706 | 695 | 61.2 |
| $\Omega_{arag}$ < 1.2 [e] | 800 | 379 | 571 | 364 | 253 | 1073 | 3440 | 984 | 78.6 |
| pH$_T$ < 7.80 [f] | 626 | 380 | 486 | 254 | 218 | 890 | 2861 | 722 | 64.3 |
| pH$_T$ < 7.70 [f] | 331 | 95 | 94 | 56 | 172 | 728 | 1476 | 286 | 33.2 |
| pH$_T$ < 7.65 [g] | 190 | 21 | 40 | 35 | 121 | 631 | 1038 | 187 | 23.3 |
| pH$_T$ < 7.52 [f] | 23 | 1 | 6 | 15 | 27 | 323 | 395 | 106 | 8.9 |
| $f$CO₂ < 400 μatm [h] | 123 | 3 | 46 | 27 | 59 | 160 | 412 | 392 | 9.3 |
| $f$CO₂ = 401–1000 μatm [i] | 674 | 426 | 675 | 451 | 140 | 517 | 2898 | 1143 | 65.1 |
| $f$CO₂ = 1001–2000 μatm [i] | 243 | 25 | 49 | 38 | 123 | 526 | 1010 | 167 | 22.7 |
| $f$CO₂ = 2001–3000 μatm [i] | 7 | — | — | — | — | 116 | 124 | 32 | 2.8 |
| $f$CO₂ >3000 μatm [i] | — | — | — | — | — | 5 | 5 | 1 | 0.1 |

[a] Total numbers of data points in the Salish cruise data package are 7526 for temperature, 7492 for oxygen, and 4449 for inorganic carbon measurements. [b] Migration blockages for adult salmonids occur at 19–23 °C, particularly in combination with oxygen levels below 3.5 mg L⁻¹ (~110 μmol kg⁻¹) to 5 mg L⁻¹ (~155 μmol kg⁻¹) (McCullough et al., 2001; U.S. Environmental Protection Agency, 2003). [c] Thresholds for sublethal to lethal hypoxia impacts range from 0.7–2.5 mg L⁻¹ for various invertebrate taxa to 1.5–4.4 mg L⁻¹ for fish (Vaquer-Sunyer and Duarte, 2008); the threshold of 2.0 mg L⁻¹ is commonly used to delineate hypoxic conditions (~62 μmol kg⁻¹ = 1.4 mL L⁻¹), with 0.7 mg L⁻¹ (~31 μmol kg⁻¹=0.5 mL L⁻¹) as a threshold used for "severe hypoxia" in the oceanographic literature (e.g., Grantham et al., 2004; Chan et al., 2008). [d] Thermodynamic saturation threshold for calcium carbonate saturation states (e.g., Zeebe and Wolf-Gladrow, 2001; Dickson et al., 2007). [e] Represents severe dissolution and growth exposure thresholds for calcifying pteropods (14 days at $\Omega_{arag}$=1.20 and 7 days at $\Omega_{arag}$=1.15, respectively; Bednaršek et al., 2019). [f] Decapod sensitivity thresholds from Bednaršek et al. (2021b) as described in text. [g] A multi-stressor vulnerability analysis specific to Dungeness crab used temperature, oxygen, and pH$_T$ thresholds of 15 °C, 62 μmol kg⁻¹, and 7.65, respectively, after testing a range of values for each parameter (Berger et al., 2021). [h] The 400 μatm $f$CO₂ value represents the approximate atmospheric CO₂ mole fraction ($x$CO₂) during 2008–2018 (the range of mean annual global marine boundary layer atmospheric $x$CO₂ across 2008–2018 is 385–408 ppm, per NOAA Global Monitoring Laboratory, 2022). Below atmospheric levels, uptake of atmospheric CO₂ by surface water air-sea exchange is favoured; above atmospheric levels, outgassing or evasion from surface waters to the atmosphere is favoured. Values of $x$CO₂ are ~2.5% higher and $p$CO₂ values are ~0.4% higher than $f$CO₂ at seawater surface atmospheric pressure and are different as a result of considering relative humidity and molecular interactions within the measured sample (Dickson et al.,



2007). [i] $fCO_2$ levels above atmospheric values have been divided into broad bins based on thresholds used in the literature to project hypercapnia impacts in fish and invertebrates (McNeil and Sasse, 2016) and behavioural to gene expression responses observed in Pacific salmon (Williams et al., 2019).

**5.2 Biogeochemical anomalies during 2014–2018**

When the atmospheric and marine heatwaves that warmed Salish Sea surface and deep waters began, a simplistic expectation

would have been that increased temperatures would drive increased rates of respiration rates in deep waters, assuming adequate supply of organic matter. If a temperature-driven increase in respiration had dominated, we would have expected to see lower-than-normal $O_2$ and $\Omega$ and higher $fCO_2$ in deep water masses during high temperature anomalies. In contrast, either surface warming or stronger-than-normal freshwater input could have limited primary production by cutting off surface nutrient supply under stronger stratification, resulting in reduced supply of organic matter and higher-than-normal $O_2$ and $\Omega$ and lower $fCO_2$

at depth. However, biogeochemical anomalies observed in this time-series did not show a simple temperature-dependent response to the heat anomaly, but rather reflected a combination of heat and river discharge influences.

**5.2.1 Hypoxia anomalies**

We know from observations and models that hypoxia often occurs in hotspots further south than Salish cruise stations on the Washington shelf during late summer (Connolly et al., 2010; Peterson et al., 2013; Siedlecki et al., 2015, 2016b), and the

resulting hypoxic water masses can be advected northward into our study area (Mickett and Newton in PSEMP Marine Waters Workgroup, 2018). Deep boundary waters anomalies observed during the heat anomaly were the opposite of the temperature-driven expectation, with a better oxygenated water column during O14 than earlier cruises this time of year (*cf.* Au08, O11, S13 in Figure 4 in Alin et al., 2023a). Fall boundary water cruises after O14 never captured hypoxic waters in JdFC and measured hypoxic conditions at one shelf station during O17, although deep waters were closer to the hypoxia threshold in

O17 and O18 than O14 and O16. The presence of deep, well-oxygenated water to greater depths than normal in boundary waters during O14 is consistent with the advection of fresher, warmed, well-oxygenated water from the NE Pacific gyre that moved on shore and dominated the upper water column during 2014–2015 (Siedlecki et al., 2016a; Peterson et al., 2017).

Hypoxic conditions were not observed in MB in this cruise time-series or in moored near-bottom time-series (Figure 9 in Alin

et al., 2023a and https://nwem.apl.washington.edu/prod_PS_ClimateTrends.shtml, respectively), likely due to the degree of mixing and shorter residence times (e.g., Babson et al., 2006; MacCready et al., 2021). Minimum deep $O_2$ values in MB during S15 and S18 were somewhat higher than during pre-MHW fall cruises, while in S16 and S17, they were slightly lower. Surface $O_2$ levels were particularly high in MB during A17 and J17, and to a lesser extent in A16 and J16 (Figure 6, Table 4, and Figure 9 in Alin et al., 2023a). In combination with observations of sustained MB phytoplankton blooms during April–August

2017 (PSEMP Marine Waters Workgroup, 2018), these high surface $O_2$ anomalies suggest that high phytoplankton biomass during spring of 2016 and spring–summer 2017 provided stronger inputs of organic matter to deep MB waters in both years, resulting in lower $O_2$ levels due to enhanced respiration at depth by the fall of each year.



In contrast, deep HC waters develop hypoxia during some years, typically in August–September in southern HC (Feely et al.,
2010; Newton et al., 2011). Hypoxic water masses were observed during O11 and S14 cruises in the process of being circulated
out of the deep southern HC basin by the fall marine intrusion (Figure 9 in Alin et al., 2023a). $O_2$ content in HC deep water
was not exceptionally low in S14 compared to earlier hypoxia years. However, during continued higher temperatures through
2015–2016, anomalously low $O_2$ conditions in deep southern HC waters were apparent as early as April of both years
(compared to A17, A18, and https://nwem.apl.washington.edu/prod_PS_ClimateTrends.shtml) and worsened considerably by
J15 and J16. The most extensive and severe hypoxic conditions captured in southern HC during the 2008–2018 Salish cruise
time-series were observed during J15 and J16 MHW cruises (Figure 6, Table 2). Unexpectedly, by S15 and S16, deep-water
$O_2$ minima were higher than in J15 and J16 as well as during S14, S17, and S18 cruises. In 2015 and 2016, early marine
intrusions flushed out deep waters and improved conditions sufficiently by September that smaller volumes of low-$O_2$ water
remained than during a relatively good pre-MHW year (*cf.* S09 in Figure 9 in Alin et al., 2023a; references in Table 1). In
contrast, S17 and S18 $O_2$ levels in deep HC water reflected hypoxic conditions equivalent to a bad pre-MHW year, in both
timing and magnitude.

### 5.2.2 Acidification anomalies

Late summer–early fall $f$CO$_2$ values in surface boundary waters ranged from <400 μatm to >800 μatm in surface waters before
the heat anomaly, with waters below 100–150 m consistently >1200 μatm (Figure S4). Comparable pre-MHW $\Omega_{arag}$ values
spanned ~1 to >2, with the lowest deep values being <0.6 (Au08, O11, S13 in Figures 7 and S4). During O14 and particularly
in O16, smaller volumes of >1200 μatm water and the deepest aragonite saturation horizon (depth where $\Omega_{arag}$ =1) were
observed in boundary waters. These observations are consistent with NE Pacific source waters with a lower respiration signal
having been advected to the Washington coast and persisting in the water column to significant depth during late 2014–2016
(*cf.* Franco et al., 2021). In contrast, the highest $f$CO$_2$ values measured in boundary waters to date were observed during O17,
from coastal to AR stations, with $f$CO$_2$ extremes of >1600 μatm and >2400 μatm measured at the surface and at depth,
respectively. Contemporaneous $\Omega_{arag}$ values were ~0.35, corresponding to $\Omega_{calc}$ values <0.6 and pH$_T$ values <7.3 (Figures S7–
S8). While these values are not unprecedented within Puget Sound, they had not been observed previously in these boundary
waters or the northern CCE.

Summer and fall cruises 2014–2018 showed most surface $f$CO$_2$ observations <600 μatm throughout MB, SS, and AR, with
spatially limited surface areas below atmospheric saturation levels (~400 μatm CO$_2$), and most of the water column having
>800 μatm $f$CO$_2$ by Septembers (Figure S5). While the lowest $\Omega_{arag}$ values typically co-occur with hypoxia, $\Omega_{arag}$ values fell
below the thermodynamic threshold ($\Omega_{arag}$=1) most times and places in Puget Sound, whereas the occurrence of hypoxic
conditions was typically limited to a few months per year in southern HC (Figures 8 and S6, Table 3). In MB, surface conditions
during S14 were similar to conditions in Au08, whereas during S15 and S16, surface $\Omega_{arag}$ conditions were somewhat lower,





but bottom waters had fewer low $\Omega_{arag}$ measurements. In contrast, MB surface waters had larger volumes of low-$CO_2$ water than usual in A17 and J17, corresponding to the particularly high $O_2$ observations and reflecting a protracted MB bloom (PSEMP Marine Waters Workgroup, 2018). These lower $f$CO$_2$ conditions were followed by unprecedented high $f$CO$_2$ and low $\Omega_{arag}$ observations throughout the water column in MB during S17 (Figures 7–8 and S5–S6, Table 2).


HC $f$CO$_2$ and $\Omega_{arag}$ temporal dynamics roughly followed the trajectory described for $O_2$. The most harmful pre-MHW conditions were observed during S09 and O11, with comparable extreme values during S14 (Figures S5–S7). More severe and widespread $f$CO$_2$ and $\Omega_{arag}$ conditions occurred during J15 and to a lesser extent J16, with $f$CO$_2$ values up to 3460 µatm, $\Omega_{arag}$ as low as 0.23 ($\Omega_{calc}$=0.36), pH down to 7.13, and $O_2$ reaching 12 µmol kg$^{-1}$ (Table 2). High $f$CO$_2$ values in S17 were similarly
severe but less widespread than in J15, and S18 conditions were as widespread but less severe than S09, which had the most widespread pre-MHW high $f$CO$_2$ values. The 12 $f$CO$_2$ observations higher than the pre-MHW record came from the J15 and S17 cruises. Relationships between $\Omega$ values and $O_2$ content in HC ($R^2 > 0.7$) indicate that aragonite and calcite saturation thresholds (i.e., $\Omega_{arag}$=1 and $\Omega_{calc}$=1) occur at $O_2$ levels of 225 µmol kg$^{-1}$ and 145 µmol kg$^{-1}$, respectively, indicating that even calcite undersaturation occurs more frequently and is more widespread than hypoxia within the southern Salish Sea. The early,
extreme J15 $f$CO$_2$ and $\Omega_{arag}$ conditions in HC were likely caused by heatwave-driven subsurface respiration. In contrast, the elevated S17 conditions appear to reflect spring–summer runoff anomalies that strengthened stratification and early seasonal blooms, which in turn would have enhanced deep respiration. Notably, the lowest minima $f$CO$_2$ across the global surface $CO_2$ observing network have been observed in HC during early spring and highest maxima in fall–winter (Alin et al. in PSEMP Marine Waters Workgroup, 2021; *cf.* Sutton et al., 2018).

**5.2.3 Implications of hydrological anomalies for future biogeochemistry in the southern Salish Sea**

While the most severe biogeochemical conditions observed in the Salish Sea through 2018 occurred in J15 and J16, coincident with the maximum expression of the MHW in Puget Sound, biogeochemical response to higher temperatures was not simple in driving the observed low $O_2$, low $\Omega_{arag}$, high $f$CO$_2$ conditions. Rather, indirect effects of increased temperature on biogeochemistry via effects of regional atmospheric warming and precipitation anomalies on watershed hydrology (e.g., river
discharge volume and timing) and circulation (stratification and deep-water renewal timing) proved to be as important in shaping the observed biogeochemical anomalies. The combination of increased atmospheric heat during most of 2013–2018, combined with record-setting precipitation anomalies and earlier snowmelt in the region, caused earlier and higher river discharges that kickstarted estuarine circulation and bottom water renewal earlier in 2015 and 2016 than typical. In this case, while HC bottom waters were more hypoxic, corrosive (low $\Omega$), and high in $CO_2$ in J15 and J16, conditions in the deep basin
had improved by S15 and S16 as a result of these early flushing events.

An independent example of the importance of hydrology to regional biogeochemical conditions came in 2017, the sole year with normal annual air temperature between 2013 and 2018. High river discharge in spring delivered nutrient and stratification





conditions suitable for unusually protracted phytoplankton blooms MB and left a fresher salinity signature that persisted until
September (Table 2). The blooms led to high surface $O_2$ and $\Omega_{arag}$ and low $fCO_2$ conditions in MB during spring–summer 2017, but were translated to unprecedented high $fCO_2$, low $\Omega_{arag}$ conditions through the water column by S17 as a result of the subsequent respiration of the high spring–summer phytoplankton biomass. However, while $fCO_2$ and $\Omega_{arag}$ experienced striking anomalies in 2017 compared to 2016 or 2018, deep MB $O_2$ levels in S17 were unexpectedly slightly higher than in S16 or S18, presumably because the water column started the growing season better oxygenated at depth in A17 than during
A16 or A18 (Figure 9 in Alin et al., 2023a). HC also experienced some of its most extreme $fCO_2$ and $\Omega_{arag}$ conditions during S17, indicating that runoff-enhanced biological processes were likely stimulated across Puget Sound basins, however, the conditions were less obviously anomalous in HC where hypoxia and extremely acidified conditions are known to recur. This interpretation is consistent with numerical simulation results indicating that freshwater inflow and estuarine exchange anomalies exerted a stronger influence on the biomass of primary producers in the Salish Sea than increased heat associated
with the 2014–2016 heat anomaly (Khangaonkar et al., 2021).

The strongest biogeochemical anomaly seen in the boundary waters was the high $fCO_2$ and low $\Omega_{arag}$ values observed throughout the SJdF water column the following month (O17). Comparing observations across October cruises indicates that $O_2$ minima were also higher in O17 than O16 or O18 in SJdF, while O17 $fCO_2$ maxima and $\Omega_{arag}$ minima were substantially
higher and lower, respectively, than observed in O16 or O18. In fact, oxygen concentrations at a southern HC location known for hypoxia were described as "the least hypoxic on record over the last several years" (Ruef et al. in PSEMP Marine Waters Workgroup, 2018). Extreme O17 acidified conditions in SJdF thus showed the same pattern as in MB of having not particularly notable $O_2$ levels compared to either O16 or O18, simultaneous with $fCO_2$ and $\Omega_{arag}$ values completely outside the range of previous observations (Table 2). Widespread low $O_2$ and $\Omega_{arag}$ conditions are known to have occurred during summer 2017 on
the Washington shelf (Olympic Coast National Marine Sanctuary and S.R. Alin, unpublished data). The driver of these shelf conditions could have been higher coastal productivity and respiration, although no pronounced coastal anomalies in temperature, salinity, or upwelling were observed during Salish cruises earlier in 2017 to support this interpretation (Figures 2–4, and S4, Table 1). However, lower coastal salinities at depth suggest a potential role for anomalous river input in coastal hypoxia in 2017 as well (Mickett and Newton in PSEMP Marine Waters Workgroup, 2018). It remains unclear whether the
strongly acidified water mass in SJdF in O17 was the water mass observed in MB and HC during S17, subsequently circulated out of PS via estuarine circulation, or the acidified shelf water mass observed in unpublished data. Either way, the acidified conditions were likely distributed throughout the SJdF water column by a downwelling wind event that occurred during or just before the O17 cruise.

Because 2017 revealed an anomaly during which $O_2$ and $CO_2$ dynamics were decoupled, manifesting as strong acidification not accompanied by particularly low oxygen or marine heatwave conditions, we dub these novel biogeochemical anomalies "$CO_2$ storms," in the sense of "carbonate weather" described by Waldbusser and Salisbury (2014). Notably, we observed these



$CO_2$ storm conditions in locations other than known acidification and hypoxia hotspots. With the higher level of background acidification in HC, what appears to be a $CO_2$ storm in other basins may manifest as less anomalous carbonate weather there, much as a rainstorm appears less unexpected in a rainforest than a desert. The background acidification and hypoxia gradient across the region—low in MB, moderate in SJdF, and high in HC—thus afforded us the opportunity to observe novel biogeochemical responses to environmental anomalies.

Collectively, the environmental anomalies of 2013–2018 yielded distinct types of biogeochemical anomalies in the Salish Sea. Marine heat anomalies may drive coupled $O_2$ and $CO_2$ system anomalies, whereas terrestrial runoff anomalies driven by atmospheric heat or precipitation anomalies can lead to decoupled $O_2$ and $CO_2$ anomalies, as shown here. To understand how future $CO_2$ storms may affect estuarine and coastal organisms and ecosystems, it is critical to have coupled $O_2$ and $CO_2$ system observations because a proxy approach to estimating carbonate chemistry from $O_2$ and physical parameters will not accurately predict $CO_2$ storm conditions (e.g., Juranek et al., 2009; Alin et al., 2012). For organisms with sensitivity to high $CO_2$ or low $\Omega$ or pH conditions, direct observation of extreme $CO_2$ events like those described here would be the only way to know that these anomalous conditions had occurred, which may in turn provide insight for interpreting observed ecological changes (e.g., changes in abundance of sensitive species or onset of a marine disease outbreak). Further, maintaining and enhancing these coupled observations will improve the ability of more complex numerical models to better differentiate and attribute the roles of and complex interactions among atmospheric, terrestrial, and marine processes in influencing estuarine and coastal acidification (Khangaonkar et al., 2021; Hunt et al., 2023).

## 6 Biological importance of understanding changing conditions in the Salish Sea and its boundary waters

The iconic marine biota of the Salish Sea—Pacific salmon, Dungeness crab, shellfish—are important resources supporting cultural well-being, livelihoods, and food security for Pacific Northwest communities. Many regional species are vulnerable to direct effects of hypoxic and acidified conditions that naturally occur in this region but are worsening due to climate change. Pink and coho salmon experience changes in their response to olfactory signals that may impair appropriate predator-avoidance behaviour during freshwater and early ocean life phases under elevated $CO_2$ conditions that are currently found in some Salish Sea marine environments (Ou et al., 2015; Williams et al., 2019). Chinook salmon are sensitive to warming temperatures and hypoxia (Crozier et al., 2019, 2021) and may be sensitive to direct ocean acidification impacts as well.

Dungeness crab are the U.S. West Coast's most economically important marine species (e.g., Alin et al., 2015) and an important recreational and tribal fishery in Puget Sound (e.g., Froehlich et al., 2017). Recent closures of the Dungeness crab fishery in HC and SS have limited regional access to this marine resource (e.g., Washington Department of Fish and Wildlife, 2020). Field studies show that early life stages experience sublethal dissolution damage to carapaces and mechanoreceptors with sensory and behavioural functions under current conditions (Bednaršek et al., 2020b). Further, Dungeness crab are



sensitive to increased temperature and declining oxygen and pH across life stages, with population-level vulnerability to projected warming, hypoxia, and acidification levels from surface to benthic habitats by the year 2100 (Hodgson et al., 2016; Berger et al., 2021). Pacific oysters, another regionally important commercial shellfish species, show a high proportion of defects in larval shell development (Waldbusser et al., 2015) in response to aragonite saturation levels currently present at most times, depths, and places in Puget Sound (Figure 8, Table 3).


Warming, hypoxia, and acidification also affect the prey and predators for these important species via trophic linkages. Euphausiids (krill) are a dominant food source for finfish and seabirds in the CCE whose larval development and survival are impaired under current pH conditions (McLaskey et al., 2016). Pteropods are another abundant prey source (Bednaršek et al., 2019 and references therein), and calcifying pteropods in the Salish Sea showed severe dissolution effects during 2014–2016

(Bednaršek et al., 2021a). Pteropods experience synergistic effects of high temperature (>11 °C), low oxygen, and low $\Omega_{arag}$ or high $p\mathrm{CO_2}$ on their abundance, shell dissolution, and oxidative stress biomarkers (Bednaršek et al., 2018, 2021a; Engström-Öst et al., 2019). Even the regional apex predator, Southern resident orca whales, whose population has been in decline for decades, may be affected if their dominant food source, Chinook salmon, declines further in abundance as a result of increasingly stressful multi-stressor conditions in the region (Hanson et al., 2021).


Many of the oceanographic variables in this observational data package passed thresholds known to be potentially harmful to regionally important species. Below we briefly describe the frequencies at which individual and multiple stressors pass known biological thresholds for salmon, crab, and pteropods in order to provide insight to marine resource managers about present-day ecosystem conditions facing Salish Sea resources. These examples of potential combined biological impacts of ocean

acidification, hypoxia, and warming in the Salish Sea illustrate how a complex ecosystem like Puget Sound manifests as a mosaic of environmental stressors occurring at different frequencies through space and time.

### 6.1 Species' thresholds to and occurrence frequencies of individual stressors

Thermal barriers to Pacific Northwest salmonid migration are known to occur at temperatures of 19–23 °C (McCullough et al., 2001). During most July 2014–2018 cruises in HC, temperatures ≥19 °C were observed in the upper 10 m of the water

column, with several exceeding 21 °C (Table 3). These observations comprise 0.5% and 0.1% of the compiled Salish cruise data product, respectively, and occurred only in HC, although July temperatures also approached stressful levels for salmon in WB. Dungeness crabs have substantial hatch mortality at temperatures >15 °C (Rasmuson, 2013). Temperatures crossed this threshold in 2.8% of observations, across all basins and dominantly in the upper 20 m, where brooding females aggregate (Pauley et al., 1989; Rasmuson, 2013).


Oxygen levels harmful to salmonids can be as high as 3.5–5 mg L$^{-1}$ (~110–155 µmol kg$^{-1}$) (e.g., Table 3 in U.S. Environmental Protection Agency, 2003 and references therein). Oxygen levels ≤155 µmol kg$^{-1}$ comprised 21.7% of Salish cruise



observations (Table 3). Dungeness crab experience feeding cessation in adults at 62 µmol kg$^{-1}$ (Rasmuson, 2013). Oxygen fell below this hypoxia threshold in 1.3% of observations, at depths of 5–335 m on both pre- and post-MHW cruises, with a

majority at HC stations and 20% in surface waters. Hypoxic bottom waters in HC have also been shown to drive Dungeness crab to shallower habitat, which may affect their catchability, predation, competition, and cannibalism, and thus potentially future population numbers (Froehlich et al., 2014, 2017). The remaining hypoxic measurements in SJdF and coastal waters (Table 2) do not represent the southern Washington coastal waters where the frequency and severity of hypoxic conditions during the upwelling season is higher (Connolly et al., 2010; Peterson et al., 2013).


As noted previously, aragonite and calcite fall below their saturation thresholds at higher $O_2$ levels, so the frequencies at which $\Omega_{arag}$ and $\Omega_{calc}$ undersaturation occur are much higher than hypoxia. $\Omega_{arag}$ was undersaturated in 61% of observations and $\Omega_{calc}$ 14% (Table 2), including hundreds of surface observations (Table 3). Biological thresholds for severe dissolution or growth in calcifying pteropods ($\Omega_{arag}$ = 1.15–1.20; Bednaršek et al., 2019) are higher than the thermodynamic threshold, thus yielding

>78% of all Salish cruise observations below this threshold, of which a third were in surface waters. While $\Omega_{arag}$ and $\Omega_{calc}$ thresholds were crossed in all seasons and basins within our study area, the threshold exceedance frequencies are much higher for calcite in HC than elsewhere, and much higher for aragonite across all basins than for calcite, temperature, or oxygen conditions becoming harmful.

Increases in $f$CO$_2$ levels may affect regionally important species through hypercapnia—the metabolic challenge of too much CO$_2$ rather than too little O$_2$. A 1000 µatm $p$CO$_2$ threshold for hypercapnia has been used for a range of fish and invertebrate studies (McNeil and Sasse 2016), although $f$CO$_2$ exposure thresholds are not well established for regionally important species. Using the 1000 µatm threshold, >25% of Salish cruise observations represent conditions potentially conducive to hypercapnia, making them more prevalent than even the 155 µmol kg$^{-1}$ oxygen threshold for salmon (*cf.* U.S. Environmental Protection

Agency, 2003; Vaquer-Sunyer and Duarte, 2008). While a discrete threshold was not identified for these changes, behavioural, neural, and gene expression responses have been observed in ocean-phase Pacific salmon between treatment $p$CO$_2$ levels of 700 µatm and 2700 µatm (Williams et al., 2019). 65% of Salish cruise CO$_2$ observations are higher than 700 µatm and 0.4% are higher than 2700 µatm, implying that ocean-phase Pacific salmon may encounter challenging CO$_2$ levels in the present-day southern Salish Sea.


A synthesis of decapod species' sensitivity to ocean acidification identified pH$_T$ thresholds of 7.80 for egg hatching success, 7.70–7.74 for adult respiration and haemolymph pH, and 7.52 for larval survival (Bednaršek et al., 2021b and references therein). Within the Salish cruise data, 9–64% of pH$_T$ observations crossed these thresholds, with pH conditions below the larval mortality threshold occupying much of the water column in HC during summer–fall (Figure S8, Table 3). Broader pH$_T$

survival threshold estimates for larval, juvenile, and adult decapod life stages span 7.40–7.80 (Bednaršek et al., 2021b); these thresholds were crossed at <10% to >60% frequencies in the Salish cruise data. A multi-stressor vulnerability analysis on



Washington and Oregon coastal Dungeness crab populations used a $pH_T$=7.65 threshold across life stages to assess exposure levels under present and future conditions (Hodgson et al., 2016; Berger et al., 2021); >23% of Salish cruise observations exceeded this threshold. Dungeness thresholds for $pH_T$ sensitivity were crossed with the highest frequency and severity in HC

(Figure S8). This chronic exposure in subsurface HC may be sublethal (Berger et al., 2021 and references therein), but the effects of acidified conditions on population distributions and crab catchability during Washington's tribal and state fisheries is currently unknown (*cf.* Froehlich et al., 2017).

## 6.2 Co-occurrence or interactions of multiple stressors in the present-day southern Salish Sea

Populations of valuable Pacific salmon and trout, including some classified as Threatened under the U.S. Endangered Species

Act, have native habitat in Puget Sound and its watersheds (NOAA Fisheries, 2022). The combination of high temperature and low oxygen can be particularly disruptive for salmonid migrations. Temperatures ≥19 °C never co-occurred in the same sample with $O_2 \leq 155$ µmol kg$^{-1}$. However, a 21.8 °C temperature occurred at 2.7 m depth with $O_2$ levels of 99.6 µmol kg$^{-1}$ and $f$CO$_2$ levels of 1437 µatm at 10.9 m at the same HC station during J18, putting three known salmon stressors in close physical proximity, with the risk of combined physiological effects and habitat compression (Table 4). Similar combinations

of temperature, oxygen, and $f$CO$_2$ conditions were also recorded during J16 and J17 in southern HC. Several salmonid runs enter a nearby river for the metabolically challenging breeding migration during summer, when these conditions formed near the river mouth (Gray, 2022). If heatwave conditions observed during summer 2015 are representative of the future marine stressor landscape, harmful levels of temperature, oxygen, and $f$CO$_2$ may be more likely to co-occur earlier in the salmon run season, rather than occurring separately, with peak temperatures in July and minimum $O_2$/peak $f$CO$_2$ values occurring in

September, as was typical during pre-MHW years.

Closer inspection of when and where harmful conditions co-develop in the region reveals that low $pH_T$ and $O_2$ co-occurred at 23 stations across 10 cruises, with four cruises and 15 of the stations sampled prior to the onset of strong, recurring atmospheric and surface water temperature anomalies beginning in 2013 (Tables 1 and 4). Of cruises with co-occurring low $O_2$ and pH, 14

stations were in HC and nine in boundary waters, and these conditions predominantly affected the bottom waters that juvenile and adult Dungeness crabs inhabit. In contrast, the combination of stressfully warm surface temperatures with low $pH_T$ near below was observed mostly after the onset of heat and other anomalies and during cruises from July into October. This combination occurred during 11 cruises at 61 stations—occupying a wider geographic distribution across PS and boundary waters—with 10 stations in August 2008 being the only cruise prior to September 2013 showing this combination of stressors

(Table 4). While these conditions are unlikely to occur in the present-day during Dungeness hatching season (winter–spring), anomalously low salinity (<15) can also prevent egg hatching, interfere with larval progression, and be lethal for adults (Pauley et al., 1989; Rasmuson, 2013). Thus, nearshore hatching habitats may thus have been affected by the major 2013–2018 precipitation and river runoff anomalies (Table 1), providing another example of how hydrological anomalies may have





profound ecosystem effects in PS, beyond effects on carbonate chemistry. We note that low salinity anomalies should also be
considered as part of the multiple stressors management landscape.

A triad of Dungeness crab stressors—high temperature (>15 °C), low $O_2$ (<62 µmol kg$^{-1}$), and low pH$_T$ (<7.65) (Berger et al., 2021)—co-occurred six times at stations in southern HC during the J15, J16, and S17 cruises (Table 4). Stressful conditions for Dungeness spanned much or all of the water column at five stations during J15 and J16 cruises, and occupied the surface
and a less extensive part of the subsurface water column during S17. This newly observed co-occurrence of all three crab stressors in Hood Canal during 2015–2017 reveals a possible future path for how the multi-stressor marine environment will evolve in this region under warmer climate conditions. Potential biological ramifications include the disappearance of suitable habitat for all life stages of Dungeness crab from some areas of the southern Salish Sea during the pelagic larval and settlement seasons. Summer crabbing seasons across 2008–2019 in HC typically spanned July–August (Washington Department of Fish
and Wildlife data), making it likely that habitat compression due to combined T, pH, and $O_2$ stress affected crab depth distributions during this time-series and may warrant management consideration. However, parsing whether the combined effects of multiple interacting stressors across crab life stages and habitats has contributed to recent PS Dungeness fisheries closures would require a more complex modelling analysis like those done in coastal waters (Hodgson et al., 2016; Berger et al., 2021).


**Table 4.** Multiple stressor events relevant to Dungeness crab, and their temporal and spatial occurrence in the Salish cruise data package. Thresholds to determine event occurrence were those used by Berger et al. (2021). Depth ranges affected by combinations of hypoxic (oxygen content <62 µmol kg$^{-1}$), low pH$_T$ (pH$_T$ <7.65), or high temperature (>15 °C) conditions are indicated in the "Depth range" columns. Cruises that occurred after the beginning of major atmospheric and surface seawater
temperature anomalies in the summer of 2013 are shaded grey.

| **Stressor** | | Number of stations[a] | | | | Depth range (dbar)[b] | |
|---|---|---|---|---|---|---|---|
| | Cruise | Boundary waters | South Sound | Whidbey Basin | Hood Canal | Boundary waters | Puget Sound |
| **pH$_T$, O$_2$** | | | | | | | |
| | Aug. 2008 | 7 | — | — | 1 | 50–335 | 85–90 |
| | Sep. 2009 | — | — | — | 1 | — | 10–45 |
| | Nov. 2010 | — | — | — | 2 | — | 5–15[c] |
| | Oct. 2011 | — | — | — | 4 | — | 5–30[c] |
| | Sep. 2013 | 1 | — | — | — | 120–330[d] | — |
| | Oct. 2014 | — | — | — | 1 | — | 10–20 |
| | Apr. 2015 | — | — | — | 2 | — | 120–150[d] |
| | Sep. 2016 | — | — | — | 1 | — | 10–15 |
| | Oct. 2017 | 1 | — | — | — | 90–100 | — |
| | Sep. 2018 | — | — | — | 2 | — | 5–55 |
| | | | | | | | |
| **T, pH$_T$** | | | | | | | |
| | Aug. 2008 | — | — | 4 | 6 | — | 5–175 |
| | Sep. 2013 | 1 | — | — | — | 120–290[d] | — |
| | Jul. 2014 | — | — | 2 | 9 | — | 20–175[e] |





| Sep. 2014 | — | — | — | 1 | — | 20–50 |
| Oct. 2014 | 2 | — | — | — | 250–325[d] | — |
| Jul. 2015 | — | — | 2 | 3 | — | 10–170 |
| Sep. 2015 | — | — | 2 | 1 | — | 5–145 |
| Jul. 2016 | — | — | — | 4 | — | 10–165 |
| Jul. 2017 | — | — | — | 7 | — | 20–170 |
| Sep. 2017 | — | 1 | 1 | 5 | — | 5–170[f] |
| Jul. 2018 | — | 1 | 3 | 6 | — | 10–165[f] |
| | | | | | | |
| **T, pH$_T$, O$_2$** | | | | | | |
| Jul. 2015 | — | — | — | 2 | — | 20–120 |
| Jul. 2016 | — | — | — | 3 | — | 20–115 |
| Sep. 2017 | — | — | — | 1 | — | 50–80 |

[a] Multiple stressors never occurred at a single station simultaneously in Main Basin or Admiralty Reach, so they are not included in this table. [b] Depth ranges affected by pH$_T$ or O$_2$ threshold exceedance are rounded to most inclusive 5 dbar intervals, although sampling resolution was sparse enough that the depth ranges affected by low pH$_T$ or O$_2$ conditions may have been larger for any given cruise. [c] The
depths at which hypoxic and low pH conditions were seen on these cruises were near-surface, not near-bottom; bottom conditions were less harmful. [d] Depth ranges affected were deeper than habitats where Dungeness crab are typically most abundant on the Washington coast (e.g., Berger et al., 2021). [e] The largest number of stations affected by two independent stressors were observed during this cruise. [f] Combined temperature and pH$_T$ stressor conditions were the most widespread across basins during these cruises.

**7 Conclusions: What we have learned about multi-stressor dynamics from the Salish cruise time-series so far**

Ocean acidification variables indicate stressful conditions throughout the region, though with substantial regional variation. Aragonite undersaturation was pervasive throughout the region, and even calcite undersaturation occurred more widely and frequently than hypoxia. $f$CO$_2$ levels were above atmospheric in >90% of observations, often by hundreds of µatms. Hypoxia was observed most frequently in southern Hood Canal and occasionally in boundary waters during 2008–2018. This seasonally resolved cruise time-series co-occurred with several major environmental anomalies, giving us the opportunity to observe the
impacts of atmospheric and marine heatwaves, precipitation, and river discharge anomalies on Salish Sea physical and biogeochemical conditions. The strongest heat anomalies manifested earlier in boundary water cruises (October 2014) but in a delayed and more protracted way within Puget Sound, spanning at least 2015–2016. Increased temperatures throughout the water column likely contributed to the most extreme O$_2$, $f$CO$_2$, and $\Omega_{arag}$ conditions seen in southern Hood Canal during July 2015 and July 2016, developing earlier in the season than the typical late-summer timing for previous hypoxia events.
However, anomalously early deep-water renewal events in Hood Canal in 2015 and 2016 resulted in late-summer conditions that resembled a less hypoxic, pre-heatwave year in deep southern Hood Canal. The effects of the heat anomaly thus underscore the importance of physical oceanographic conditions (i.e., temperature, stratification, and circulation timing) in setting the stage for the severity and duration of ocean acidification and hypoxia in local hotspots.

Both seasonal and spatial variation was strong. The decade-long Salish cruise time-series illuminated several differences in the seasonality across parameters and across Puget Sound and its boundary waters. Some parameters peaked at the surface during summer (temperature, $\Omega_{arag}$), with others rising (salinity, $f$CO$_2$) or falling (O$_2$) from spring to fall, while all parameters





showed more monotonic seasonal progression at depth. Within Puget Sound, the largest overall variability was observed in the most stratified basins. Notably, deep-water seasonality in Hood Canal was different from other basins in that seasonal $O_2$ lows and highs co-occurred with temperature and salinity minima and maxima, whereas in other basins, salinity and temperature were inversely correlated with $O_2$ content. The lags likely result from the combination of more complex deep-water renewal and earlier spring blooms in Hood Canal, which appear to decouple its physical and biogeochemical seasonality relative to other deep basins and may predispose it to more hypoxic, acidified conditions. To the extent that climate change alters the coupling between physical and biogeochemical seasonality in other regional waters, changes in the frequency, severity, and duration of ocean acidification and hypoxia may be expected.

During fall 2017, we observed a novel carbonate system anomaly in Puget Sound and boundary waters. This "$CO_2$ storm" was characterized by unprecedented high $f\mathrm{CO_2}$ and low $\Omega_{arag}$ values, which crossed sensitivity thresholds for regionally important species. This extreme event occurred independently of heatwave or particularly low conditions and instead appears to have resulted from major river discharge anomalies reflected by low salinity anomalies observed earlier in the year. The $CO_2$ storm was most obvious in basins with less acidified baseline conditions, but was still detectable in the most acidified basin, Hood Canal. These observations showed decoupling of carbonate chemistry from that of oxygen, which was unusual in this time series. This result underscores the need for on-going seasonal monitoring of both carbon and oxygen across Salish Sea basins, which showed variable responses to anomalies in physical ocean conditions, river input, and local weather. The Salish cruise time-series has thus illustrated how the arc of major environmental anomalies and their ecological impacts depends on the biogeochemical metric, species of interest, and baseline conditions of the basins in which they occur.

The frequencies at which Salish cruise observations crossed known or preliminary species sensitivity thresholds illustrates the relative risk landscape of temperature, hypoxia, and acidification anomalies in the southern Salish Sea in the present-day. Interactions between marine heat and other environmental anomalies during the 2014–2018 cruises revealed how multiple stressors can combine to present potential migration, survival, or physiological challenges to key regional species. Collectively, the occurrence frequencies of these combinations of stressors for Dungeness crab, salmon, and pteropods in the Salish cruise data package illustrate how increasingly frequent and severe marine and atmospheric heatwaves may alter the future co-occurrence of multiple stressors in the southern Salish Sea. Specifically, while low $pH_T$ and $O_2$ co-occurred regularly before the heat anomalies in mostly subsurface Hood Canal and boundary waters, the co-occurrence of high surface temperature and low pH conditions appeared to increase sharply across the marine heatwave, with a broader spatial distribution than where low pH and $O_2$ previously co-occurred. Future changes in the seasonality of when harmful conditions develop and co-occur may cause sensitive species' thresholds to be crossed more frequently due to changes in timing even without changes in severity. The novel forms and combinations of extreme events observed in the 2014–2018 Salish cruise time-series have thus provided insight into the potential evolution of the future marine stressor risk landscape in the Salish Sea.





## 8 Data availability and data use

This analysis is focused on a subset of the Salish cruise data product archived in Alin et al. (2022, https://doi.org/10.25921/zgk5-ep63) and described in Alin et al. (2023a). To facilitate use of the combined multi-stressor data product at the core of this paper, we created a novel data product that includes only the highest-quality measured parameters,
along with the calculated $CO_2$ system parameters $pH_T$, $fCO_2$, $pCO_2$, $\Omega_{arag}$, and $\Omega_{calc}$, which were calculated using two sets of dissociation constants as described in Section 3.2; this multi-stressor data product is archived at Alin et al. (2023b, https://www.ncei.noaa.gov/data/oceans/ncei/ocads/metadata/0283266.html). The data (filename= "SalishCruises_2008to2018_MeasCalcParams_NCEIdataProduct_09262023.csv") and metadata (filename= "SalishCruise_2008to2018_MeasCalcParams_metadata_09262023.xlsx") can be accessed by clicking the "Download Data"
button and downloading the files from the Index page.

**Author contributions**

SA led analysis of inorganic carbon samples, assembly and analysis of data and metadata, interpretation of data analyses, and manuscript drafting. JN led organization and execution of all cruises, oxygen and nutrient measurements, and provided input on data analysis and interpretations at all stages of the work. RF contributed to the development and implementation of this
project and the writing of the manuscript. SS was our research partner on many projects over the duration of these cruises and contributed to understanding the physical and biogeochemical dynamics in the region. DG prepared transect profile graphics and made major contributions to data wrangling. All authors contributed to editing the manuscript.

**Competing interests.** The authors declare they have no competing interests.

**Acknowledgments**

We acknowledge that the land our laboratories are located on has been the home of Coast Salish people since time immemorial and that our study area encompasses the traditional and ancestral waters of the Coast Salish peoples and the Coastal Treaty Tribes of Washington. We thank the leadership and technical staff of numerous research and monitoring organizations that made the southern Salish Sea time series of cruise observations available for this analysis, as detailed in the companion article submitted to *Earth System Science Data*, with particular gratitude to the Washington Ocean Acidification Center for stable
funding and sampling schedules since 2014 that allowed the seasonal characterization to be done. SRA and RAF are deeply grateful to NOAA Pacific Marine Environmental Laboratory for supporting their sustained effort on this project. Li-Qing Jiang and Alex Kozyr of NOAA National Centers for Environmental Information provided invaluable feedback on the manuscript and assistance with archival of the data products associated with these papers, respectively. We thank Don Velasquez of the Washington Department of Fish and Wildlife for providing information about Hood Canal Dungeness crab fisheries dates for



the study period and Cindy Gray and Seth Book of the Skokomish Tribe Department of Natural Resources for information about Hood Canal salmon species migration timing.  This is PMEL contribution number 5297.

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
