# Peer review of "Seasonality and response of ocean acidification and hypoxia to major environmental anomalies in the southern Salish Sea, North America (2014–2018)"

_Biogeosciences, 2023_

## Author Response (AR1)

Dear editorial team:

We appreciate the opportunity to improve the manuscript based on the input of reviewers 1 and 2, whose comments were very helpful for indicating where we lacked adequate clarity or detail. We believe we have addressed their comments appropriately, as detailed below, and note that we made a few minor wording changes regarding circulation in the region based on publications that were not yet published when we submitted or were missed by the lead author.

We look forward to hearing back from you.

Best,
Simone and co-authors
* * *
We thank anonymous reviewer 1 for providing many helpful comments and constructive suggestions for clarifying our manuscript for a broader audience.

**REVIEWER #1**

General comments:

Coastal ecosystems in the North Pacific Ocean, specifically the southern Salish Sea, face heightened vulnerability to ocean acidification, hypoxia, and marine heatwaves due to natural and human-induced factors. A detailed analysis of a seasonal cruise time series in this region reveals varying patterns in physical and biogeochemical parameters, with some peaking in summer surface waters while others change progressively from spring to fall. Unusual environmental anomalies, including a significant temperature anomaly in 2014 and a unique "CO2 storm" in 2017, driven by anomalous river discharge, highlight the complex interplay of multiple stressors with potential implications for regional ecosystems and key species in the future. The document is well written, but some clarifications are needed to improve readers' understanding:

- Section 3: the methodology is weak, and more detail is needed on the methods used to correct the data set, while the assessment of uncertainties for the variables is not indicated and is clearly lacking. This is particularly the case for salinity and O2 concentrations, which are sensitive to the effects of biofouling.

We added three sentences to section 3.1 to give a brief synopsis of the variables measured, their respective uncertainties, and where the detailed methods for the data collection and quality control were described. Detailed methods and uncertainties are in the Alin et al 2023 ESSD companion paper, and quality control methods are from Jiang et al 2021 ESSD paper on Coastal Data Analysis Product in North America (CODAP-NA). We believe the language added to this section should

address the reviewer's concern about lack of details of the methodology, which are in the companion paper. We simply failed to spell this out clearly enough in the submitted version.

However, there should be no biofouling concerns regarding salinity and $O_2$ measurements, as these measurements are solely collected from ship-based CTD casts. The sensors were never in the water more than an hour at a time, deployed at a few dozen stations over the span of a few days to a week at most, on each cruise. Bottle/Winkler $O_2$ samples were collected immediately after casts were conducted and were preserved and analyzed on board the ship within a few days of collection; for all cruises, Winkler $O_2$ values were compared to CTD sensor data after processing. On some cruises, there were dual $O_2$ and/or salinity sensors to compare to each other (no salinity bottle samples). While the comparisons between dual sensors are not described in the ESSD paper, the bottle vs. CTD $O_2$ comparisons are described and there are plots included, none of which suggest biofouling. For salinity, the only major difference we ever noted was when calibration coefficients were incorrect (e.g., giving salinities of 35 in this region, which have never been observed); these out-of-range values spurred us to realize that calibration coefficients had been incorrectly loaded. There are a handful of bad salinity values, for reasons unknown, that were apparent in plots of alkalinity vs salinity in the ESSD paper; those data were not used in this analysis.

- Section 5.2.3 Provide a brief explanation or context for the term "CO2 storm" to help readers understand its significance and how it differs from typical conditions. This would improve accessibility for a broader audience.

We added some more detailed language to clarify that the observed "$CO_2$ storm" was the *presence of particularly high $fCO_2$ (and low $\Omega$ and pH)* that was not accompanied by particularly low $O_2$ or marine heatwave conditions **in places where these conditions have not been observed to occur in the past**. (The information added to the text is paraphrased by the words bounded by asterisks above.)

In this section, the authors do not discuss the evolution of CO2 sinks and sources driven by the temperature change, biomass production and atmospheric forcing. According to the evolution described above, what would be the impact on CO2 fluxes in terms of sources/sinks in these coastal waters?

While the evolution of $CO_2$ sources and sinks is of course germane to temperature change, biomass production, and atmospheric forcing, this paper was intended to focus more on the evolution of oceanographic conditions and their effects on sensitive marine species, during a period with several marked environmental anomalies. This specific section (5.2.3) is specifically about the effects of hydrological events on biogeochemical conditions in Puget Sound, which has not received adequate attention previously. Thus, we are reluctant to dilute the focus on the role of hydrological events here by discussing $CO_2$ fluxes, sources, and sinks, which are surface-focused phenomena, in this manuscript where we are looking across the water column at biogeochemical evolution in response to extreme events. We instead propose to work with one of our co-authors, who has a manuscript in preparation about the evolution of air-sea fluxes through seasonal cycles in the Salish Sea, to include consideration of the role of extreme events in combination with these fundamental regional

drivers (i.e., temperature change, biomass production and atmospheric forcing) on air-sea flux. We appreciate the reviewer's suggestion and hope to better address it in this additional paper that is currently in draft stage.

- The conclusion is too brief. How to optimize the observations to better observe and understand the evolution of the coastal marine ecosystems in the North Pacific towards acidification and O2 decrease ? What will be the next challenges ? Which data or tools will be important to include for the next study  (eg. better obs. space/time frequencies ? etc…). Consider adding a sentence or two in the conclusion to summarize the key implications or recommendations based on the findings. This would provide a more comprehensive and conclusive end to the abstract.

We really appreciate these comments and have added a few sentences to the conclusion to underscore some of the points suggested.

Minor comments:

- Line 83: explain what do you mean by the "time of emergence"

We have added text in this line to explain more fully what "time of emergence" means.

- Figure 1: there is an error of typology

Since the reviewer did not provide details, we are guessing they are referring to the mooring name, which has non-standard symbols in it. We added a few words to the caption to explain that this is a name from a North American tribal language.

- Figures 3 and 4: The raincloud plots are not easy to read. The Fig S4 to S6 are more useful. The authors should explain more the signification the raincloud plots in the text or in the legend. A map with stations acronyms like in Fig S5 would be very useful to understand the trends. Authors should insert a map in the Figures 3 to 4. In these figures MB and SS could be combined as it is shown in Fig. S5 to clarify the figure

We thank the reviewer for their helpful feedback about the raincloud plots, and we also greatly appreciate the depth transect visual format for displaying our oceanographic data in Figures S4–S8. However, we had previously received feedback from interdisciplinary colleagues (i.e., non-oceanographers) that those figures were too dense/complex to be readily interpretable by non-specialist audiences. They specifically requested to also see some sort of visual statistical summary of the data. We decided on raincloud plots because they reflect the seasonal, spatial, and depth variability across parameters and across this diverse region.  We are aware that they are still complex, and that it is not easy to pick out details. Summarizing this volume of data efficiently, clearly, and intuitively is a major challenge; we tried many other types of plots on the way to deciding on the raincloud plots!  We also provided the ranges of values for each parameter in Table

2, in an effort to ensure all of our end users can find the information they most need. In the companion ESSD paper, we have ALSO provided the detailed depth transects preferred by reviewer 1 for all parameters not included in the supplemental material for this paper. To address reviewer 1's concern, we have added an explanation to section 3.4 for readers that prefer that form of data visualization, indicating that depth transect plots can be found in the supplemental material of this article and in the companion article and its supplemental material.

- Oct 2017 is a very particular month but not shown in Fig. S5 to S8, why ?

Oct 2017 was the cruise month in the Sound to Sea plots (Figures 7A, 8A, and S4) that was very anomalous, but the corresponding measurements in Puget Sound's Main Basin (Figures 7B, 8B, and S5–S8) were in Sept 2017. We revised the text in the same paragraph where we elaborated on what we meant by "$CO_2$ storm" to clarify this distinction.

- The place of figure 5 is bizarre and only discussed in section 4.3

We've added a few words to section 4.1, where this figure was first discussed, to indicate that the purpose of it was to facilitate summarizing across years, regions, and parameters. The figure was also discussed in sections 4.2 and 4.3.

- Line 828: what are the numerical models discussed here ?

We added language to clarify that these are 3D physical-biogeochemical ocean models.
* * *
We appreciate the helpful and complementary comments from anonymous reviewer 2 that will help further clarify our manuscript for a broader audience.

**REVIEWER #2**

**Review of "Seasonality and response of ocean acidification and hypoxia to major environmental anomalies in the southern Salish Sea, North America (2014–2018)" By Alin et al.**

**General comments**

The research presented in this paper holds significance due to the influence of variations in biogeochemical parameters, such as dissolved oxygen, fCO2, pH, and carbonate minerals' saturation, on the ecosystems' well-being amid the looming challenges of climate change. It establishes connections between environmental and biogeochemical anomalies. The study focuses on the southern Salish Sea, where intricate interactions between anomalies in physical ocean conditions, river input, and

local weather significantly influence the severity and duration of ocean acidification and hypoxia in local hotspots in this complex region. The Salish cruise time series data effectively demonstrates how the progression of significant environmental anomalies and their ecological repercussions rely on the specific biogeochemical metrics, the species under consideration and the baseline conditions of the basins in which they unfold.

Considering the comments already sent by Anonymous Referee #1, here are my specific comments:

**Major comments**

The flow of information in the 2.1 Geographic setting section is challenging to follow; I recommend following a structure similar to that of the article by Alin et al., 2023a, cited in the references section.

Thank you for this comment. We've reorganized section 2.1 as suggested, such that we believe it both flows better and follows a more similar structure to the comparable section in our ESSD companion paper.

**Minor comments**

Line 25: I would also include the utilized study period of the time series in the abstract.

Done.

Line 115: Refer to the Figure 1.

Done.

Line 117: For a better understanding, it would be advisable to refer here to Figure 1 or indicate the location of the Fraser River, as shown later in Line 126.

Done.

Line 121: The abbreviature PS appears before it is explained in Line 122.

We added the abbreviation in location earlier than the first use of PS, as we should have. There was some reorganization of this section, so this line now appears earlier in section 2.1.

Line 125: You could add the abbreviation of the different Puget Sound basins here instead of that in Line 131 (MB), MB repeated in 133 and the remaining in 134.

Good idea. We have added them to the suggested line (which now occurs earlier). We do repeat the full basin names on occasion throughout the paper, as we transition from discussing one basin to another, to help readers less familiar with this region follow the flow.

Figure 1: I recommend including a legend with the abbreviation of the different transects and the associated data point types for a more straightforward interpretation.

We revised Figure 1 so that the legend is incorporated inside of the figure and have adjusted the caption text accordingly. A good suggestion.

Line 210: Although it is assumed that the description of the variables analysis methodology, like DIC and TA, is included in the references related to the database, it would be helpful for the reader to add a sentence here indicating the references where they are found again.

We have added three sentences to section 3.1 to give a brief synopsis of the variables measured, their respective uncertainties, and where the detailed methods for the data collection and quality control were described (Alin et al 2023 ESSD companion paper and Jiang et al 2021 ESSD paper on Coastal Data Analysis Product in North America (CODAP-NA), respectively). This addition was in response to a comment by reviewer 1, but if we understood reviewer 2's comment here, I believe these changes addressed this concern as well.

---

## Author Response (AR2)

Dear Fréderic and editorial team,

We are delighted by this news! We have replied briefly to your suggestions below. All replies in blue below.

We note that to prepare final figure files, we made superficial edits to make the labels more attractive on figures 2-9; no data or other substantive information was changed in so doing. Fig 1 is exactly as resubmitted in the previous rounds. For figures S5-S7, we also edited scale bar labels to be more consistent with abbreviations in the text.

We also made minor edits to correct figure references to Copernicus style and added a brief financial support statement to the end, as I think is now required.

Please let us know if you have any questions. Thank you for your time and attention to our work.

All the best,
Simone and co-authors

I am happy to accept your revised manuscript following very few technical corrections which I would like you to consider.

L35: consistently had

Changed.

L44: I do not really understand this sentence, are we talking about present day or risks for the future (I would remove "in the future" at the end of the sentence but am not sure this is what you meant)

We see your point and appreciate the opportunity to clarify. While we did intend to say "in the future," the reality is that those impacts may also be occurring now. So we removed "in the future," because this way it may be read to apply to now. The implication that it is even more likely to do so in the future should also be clear.

L84: the projected (remove future) time

Done.

L251: define CTD

Done.

Section 3.2: I appreciate the amount of details given here but I wonder if this is important for most readers, could you consider moving at least some of the information given here to a supplementary material? Also, did you use the Error function of seacarb for the uncertainty propagation? If so, it should be mentioned as well.

Good suggestion. We shortened this section from 56 lines to 30 lines by moving many of the details to a new Supplemental Material section. We did not use the Error function of seacarb. These are just the estimated errors in Orr et al. 2018; we have edited the sentence to clarify this.

L275: Why is it important to mention a regression determination coeff for 2 variables that are calculated using similar dissociation constants? Maybe I am missing something here.

Fair point. We have removed the coefficient of determination wording.

L337: prepared in Surfer and can be found

Done.

L338: Comparable plots of temperature, salinity, as well as oxygen, DIC, TA and nutrient content

Corrected to "Comparable plots of temperature and salinity as well as oxygen, DIC, TA, and nutrient content..."

Figure 5: I agree with R1 that the position of Figure 5 is odd, I see that you cite it now regarding S and T in section 4.1. Perhaps a better solution would be to divide this figure in 2 and present O2, fCO2 and Omega panels closer to section 4.3. Also remove PSU from Y-axis label.

Thank you for catching the errant PSU and for the idea to split the figure in two. That seems like the solution we did not come upon prior to resubmission. We have done so now and updated all figure references in the text.

L471: median values from April to September

Done.